# Coastal Waterfront Vibrancy: An Exploration from the Perspective of Quantitative Urban Morphology

**Lung Shih Huang, Yun Han * and Yu Ye ***

College of Architecture and Urban Planning, Joint International Research Lab of Eco-Urban Design,
Tongji University, Shanghai 200092, China
* Correspondence: yunhan1124@163.com (Y.H.); yye@tongji.edu.cn (Y.Y.);
  Tel.: +86-199-2127-6317 (Y.H.); +86-182-1726-8257 (Y.Y.)

**Abstract:** Contemporary urban design, requiring a deep understanding of urban form and its performance, has recently shifted its focus on the vibrancy of waterfronts in coastal cities. Based on analytical methods of quantitative urban morphology, this study aims to explore the common morphological features of waterfronts with high urban vibrancy. We selected vibrant waterfront cases from different countries as the benchmark and collected the multi-sourced urban data. The quantitative analysis extracts the common morphological characteristics of vibrant waterfront by calculating the range of those indicators in different cases. The results indicate that those successful waterfronts comprise compact street networks and are mostly dominated by building types favorable for urban vibrancy. They possess high development intensity and mixed functions. Consequently, the compact urban form and dense-mixed land use are recommended for developing vital waterfronts. Moreover, considering the problematic waterfront area of the Jinshatan area in Yantai, quantitative urban morphology methods can be adopted to develop precise urban design guidance for vibrancy-oriented design practice. This study, thus, provides comprehensive insights for shaping the vibrancy of the waterfronts in coastal cities.

**Keywords:** coastal waterfront; urban vibrancy; quantitative urban morphology; multi-sourced urban data

## 1. Background

### 1.1. The Significance of a Vibrant Waterfront in Coastal Cities

The vibrant waterfronts are often the core areas of coastal cities, attracting diverse activities and representing the image of the city. Thus, they tend to have important historical value and potential for tourism development, benefiting economic development to some extent. The mid-20th century witnessed a decline in coastal manufacturing industries due to the transfer of international labor division, with industries shifting from developed to developing countries. However, some coastal cities, such as Baltimore's Inner Harbor [1], Boston's Quincy Market [2], and Sydney's Darling Harbor, have revitalized their waterfront spaces through urban regeneration. In this context, there is a broad consensus on integrating waterfront areas into coastal zone management through urban design [3], revitalizing the waterfront while ensuring sustainable development, and creating livable spaces [4]. Accordingly, quantitative urban morphology has garnered considerable attention from researchers and practitioners as an effective urban design tool.

### 1.2. Urban Vibrancy as an Important Indicator of a Living Waterfront Space

Many coastal waterfronts need to be shaped into livable places for human life. Urban vibrancy denotes the intangible spatial quality created by diverse social activities [5], thus serving as a key indicator of the livability of the waterfronts. The various amenities create the conditions for attracting social activities continuously throughout the day [6].

Numerous empirical studies have demonstrated that diverse social activities can promote social interactions and enhance social cohesion [7]. Additionally, creating urban vibrancy serves to enhance the gateway function of waterfront areas in coastal cities [8]. Moreover, vibrancy intensity characterizes the concentration of social activities, which is an essential indicator of the efficiency of waterfront space utilization [9]. Consequently, urban vibrancy is a key indicator of spatial quality, related to the livability, attractiveness, and sustainability of waterfronts [10].

### 1.3. Research Objectives and Significance

According to the above introduction, it is obvious that enhancing the vibrancy of human habitation is an effective way to achieve holistic and sustainable development. Although many studies have proved the correlation between urban morphological characteristics and urban vibrancy [5,6], it still lacks a systematic understanding of the common characteristics of those urban form features of the vibrant waterfronts. In line with this gap, there is still a lack of an operational path to guide the arrangement of urban form indicators that is aimed at developing vibrancy.

In response, this study targets the human-settled coastal waterfronts that are faced with the urgent need to revive vibrancy. A series of quantitative morphological indicators is demonstrated according to the literature review in Section 2. Based on those indicators, this study selects the benchmark and problematic cases and proposes a systematic path to support the vibrancy-oriented urban design practice for coastal waterfronts in Section 3. Then, the morphological indicators summarize the common characteristics of vibrant waterfronts and show specific guidance for the problematic area through the quantitative urban morphology methods in Section 4. Finally, the contribution and deficiencies of this study are further discussed and concluded in Section 5.

## 2. Literature Review
### 2.1. The Vibrancy of Coastal Waterfronts and Their Relationship with Urban Form

In urban design practice, how to enhance spatial quality through intervention in urban form has always been a central concern [11]. Therefore, creating vibrancy from the perspective of spatial form has always been the focus of the coastal waterfront.

The relationship between spatial form and urban vibrancy has been demonstrated in many previous studies. From the qualitative perspective, researchers pointed out that urban vibrancy consists of an integrated urban spatial quality of social activity and is associated with many spatial forms tightly [5,6,12,13]. According to Jacob's classical urban design theory, the vibrancy-related urban form features include four aspects, the mix of building ages, multiple building functions, building density, and street accessibility. Among them, the concentration of buildings will facilitate social interaction, and the diverse functions and chronology of buildings will attract a diverse range of people to gather in public spaces. Moreover, a high level of spatial pedestrian accessibility will promote walking trips for city dwellers and enhance social interaction activities in public spaces [12,13]. In terms of the quantitative aspects, researchers further verified the contribution of spatial elements to vibrancy, such as locations [7], small blocks [5,12,14,15], dense street networks, high intensity, and mixed land use [16]. Some researchers also proved the effectiveness of the accessibility to certain functions, such as small catering facilities [17] and subway stations [18]. In general, there is a broad consensus on the spatial characteristics that are conducive to vibrancy, such as street intersection density, building density, block size, and land use diversity.

### 2.2. Quantitative Urban Morphology and Its New Potentials

With the progress in urban analytics methodology over the past decades, the methods to describe, classify, and represent the spatial and socio-economic performance of urban form gradually become mature, known as quantitative urban morphology [19]. Urban morphology was defined by Conzen's theory originally, including street configuration,

development intensity, typology, and land use aspects [20]. Quantitative urban morphology uses new analytical tools to quantify the urban form and extract the indicators from the urban morphology.

Recently, the emergence of multi-sourced urban data has opened up new opportunities for quantitative urban morphology as an approach to urban design analysis. First, geospatial data make it possible to refine the measurement of urban spatial form on a large scale [21]. Second, the functional service facilities data help to quantify urban land use features in detail, such as the diversity and density of different urban facilities [17,22]. More importantly, the quantitative urban analytics tools make it possible for automatic urban analysis [16,23]. For example, based on the space syntax, the accessibility of streets could be predicted accurately through the spatial analysis network plugin on the software of ArcGIS [21]. Moreover, Spacematrix, invented by Berghauser Pont, is a useful technique that can classify various building types quantitatively with a set of morphological indicators, e.g., GSI and average floors [24]. Overall, the urban morphology methods assisted by big data and new analytical tools will bring new insights into the urban design of developing vibrancy under the specific environmental context.

*2.3. Research Gap and the Purpose of This Study*

Based on the above literature review, two research gaps could be concluded. First, although many empirical studies have been conducted to discover the elements of urban form that are closely related to vibrancy, few have proposed a systematic path to guide the formation of vibrancy from an urban morphology perspective. Second, although the correlation coefficients between key morphological features and urban vibrancy have been analyzed, detailed thresholds of these key elements of urban form are often missed. An understanding of the valid range of thresholds would help to provide precise guidance on urban design practice.

To address the above gaps, the quantitative morphological indicators could be applied to guide the vibrancy-making process of waterfronts in coastal cities. First, the successful benchmark waterfronts in different coastal cities are selected, and the common urban form characteristics of those benchmarks are summarized. Second, this study takes the problematic coastal waterfront area as an example, exploring the specific guidance of spatial elements from the perspective of quantitative urban morphology.

## 3. Research Method

Firstly, waterfronts famous for their vibrancy were selected as the benchmark cases and the Jinshatan waterfront area in Yantai was selected as a comparative case. Secondly, we identified the urban morphological indicators in three dimensions, including street network configuration, land use, typology, and development intensity. Detailed sources for the three dimensions can be found in Section 3.2. Subsequently, we summarized the overall range of values for each morphological indicator of those benchmark cases. Then, the urban form indicators of benchmark cases were compared to the problematic case. The analysis results would finally help us to propose the targeted urban form guidance and control strategy for a problematic case. The analysis framework is shown in Figure 1.

*3.1. The Case Selection and the Research Scope Definition*

3.1.1. The Benchmark Cases: The Vibrant Coastal Waterfront Areas

The successful waterfront areas were selected according to the following specific criteria: (a) they feature important coastlines near city centers or urban areas; (b) they contain abundant good water and green resources; (c) they are known internationally for their urban vibrancy, providing abundant leisure lives that attract urban dwellers; and (d) they have long histories, sufficient development, and well-complete urban form.

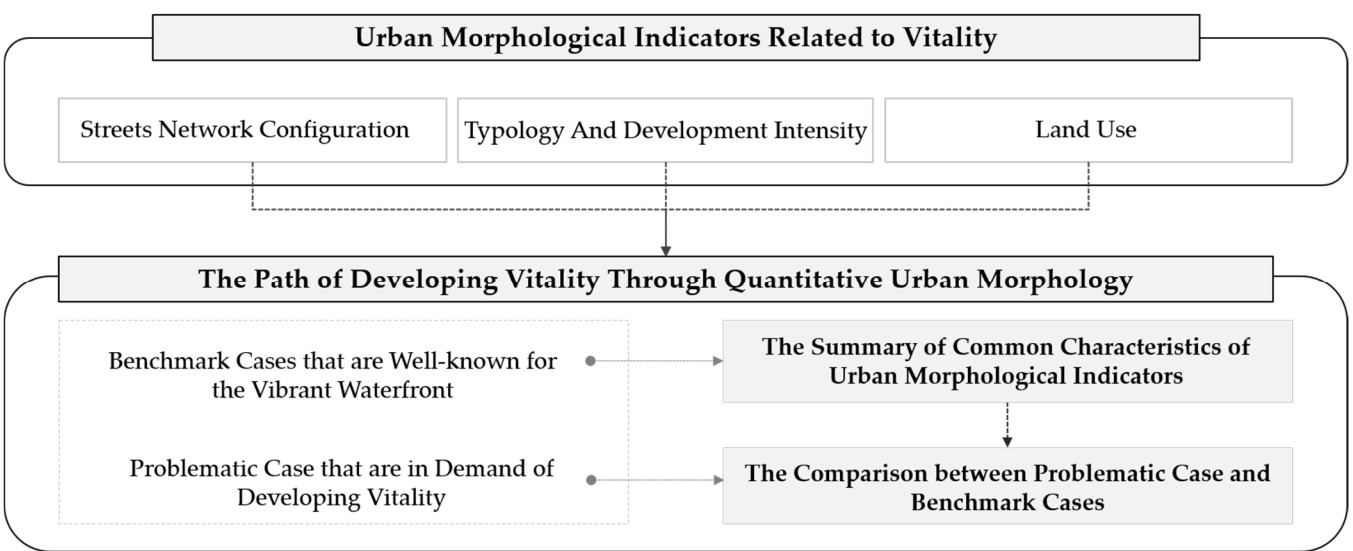

**Figure 1.** The analytical framework.

Accordingly, through the world's leading travel website, tripadvisor.com (accessed on 12 September 2022), we found some famous vibrant waterfronts located in the USA and Europe. Then, we selected the study areas conforming to the conditions above as benchmark cases [25]. The study area of each case is from the coastline to the hinterland one kilometer away (a ten-minute walking distance), with an area of 3 to 5 square kilometers (Figure 2).

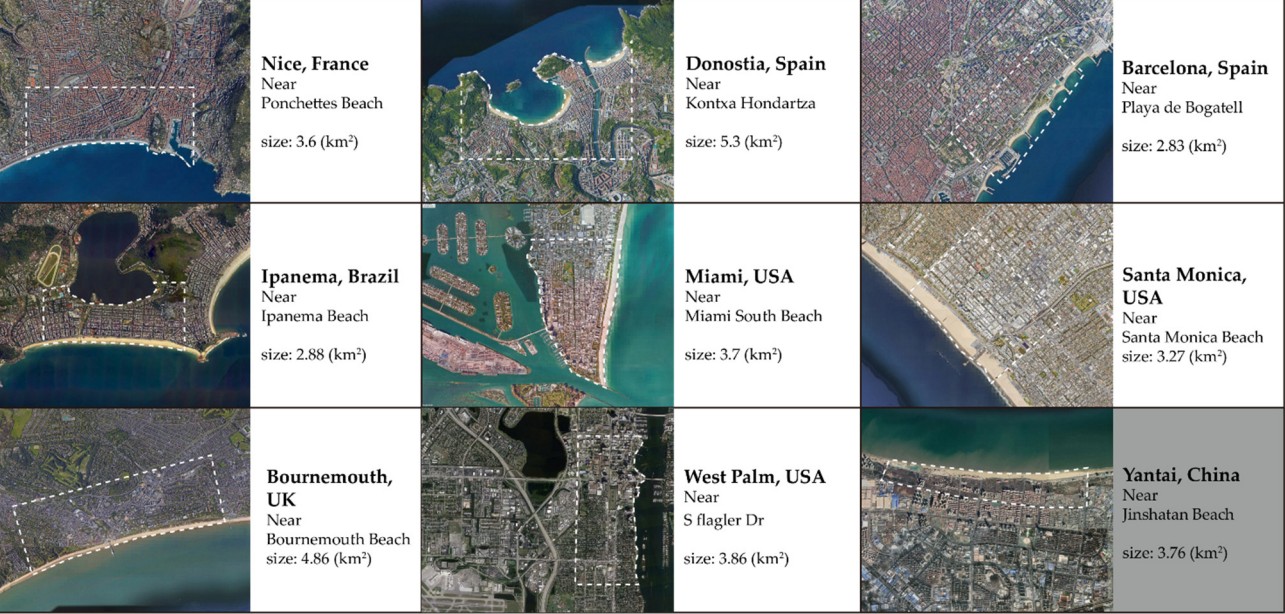

**Figure 2.** The waterfront areas selected by the vibrant coastal cases and the Jinshatan area.

The values of the indicators from these cases could guide the form of other areas with similar environmental backgrounds but lower vibrancy. Although the cases we choose come from different places of the world and even have different urban morphology in some aspects, their many important features above are the same and are all famous for their urban vibrancy. Consequently, we assume that the famous waterfront urban areas might have common ranges of indicators in urban morphology. Their common ranges might be able to form advice for the Jinshatan area, a waterfront area with a mixed function but lacking urban vibrancy.

### 3.1.2. The Problematic Examples: A Coastal Waterfront Area in Yantai

Jinshatan is a provincial tourist resort located in the Yantai Economic and Technological Development Zone, in the coastal belt of northern China. It has more than 10 km of sandy beaches. It is also a comprehensive area dominated by the tourism leisure industry and is adjacent to a residential area. The Jinshatan area has numerous geographical and environmental advantages. However, based on the relevant reviews on the website of Tripadvisor.com [25], it can be found that, compared to other famous waterfront areas in China, this area has a lower level of tourism prosperity.

### 3.2. Morphological Indicators Aimed at Developing Vibrancy of the Coastal Waterfronts

Starting from the "Anglo-Germanic Historico-geographers" and "town-plan analysis" research methods proposed by Conzen [20], the urban form can be disassembled into town planning, typology models, and land use models. Town planning can be divided into street organizations, plots, blocks, and buildings. According to the definitions mentioned above, to summarize the relevant features, the quantitative urban morphology can be divided into three dimensions, as shown in Table 1, including street network configuration, typology and development intensity, and land use. The meaning of each indicator has also been explained below.

After disassembling the urban form into three types, according to the classical theory of urban vibrancy and urban form [5,6,26–28], the indicators affecting urban vibrancy can be extracted from each dimension. Specifically, road density, road area density, street accessibility, block size, and road intersection density are extracted from street network configuration. Generally, compact road networks, short blocks, and high-accessibility areas contribute to urban vibrancy [29–35]. From the dimension of typology and development intensity, building types, floor area ratio, ground space index, open space ratio, and average building floors can be extracted. Usually, areas with high development intensity may contribute to urban vibrancy [36–41]. Furthermore, several kinds of building are conducive to urban vibrancy. In addition, the average mixed-use index, the proportion of functions, and the density of facilities are extracted from the land use dimension. Empirical studies show that the higher the degree of mixing and density of facilities in an area, the more conducive it is to the vibrancy of the city [42–44]. The function proportion also affects urban vibrancy [45]. Table 1 lists urban morphological types, indicators, definitions, and formulas in detail.

The data utilized in this study were obtained from open-source data websites, such as Openstreetmap.org, and Python programing language was used to request Pois from Google Map API. The data were integrated and calculated using ArcGIS software. The betweenness indicator was developed using the sDNA, an ArcGIS plug-in.

To provide insight into the urban design that is often practiced based on block scale, we have used the block as the calculation unit for each indicator, except the indicator of betweenness. There are two reasons for this consideration. First, the block is widely regarded as an operation spatial unit when designers intervene in the urban form [46]. Second, the effectiveness of block has been verified by many empirical studies [14,15,34]. As to the indicators of road density, road area density, and road intersection density, we calculated their values by the road within the walking distance of 300 m [47] for each block. However, the road segment has been used as the calculation unit for the indicator of betweenness. That is because this indicator mainly reflects the street network structure; thus, we want to know which roads have higher values and have more opportunities to be the regional or city centers through the betweenness.

**Table 1.** The morphological indicators.

| Morphology | Indicators | Definition | Formulas |
|---|---|---|---|
| Streets Network Configuration | Road density [29,30] | The total length of the street network divided by the area | R_d = L/A<br>R_d: The road density, L: Total length of roads in the area (km), A: The land area |
| | Road area density [29,30] | The sum of the street network area divided by the area | R_ad = R_a/A<br>R_ad: The road area density, R_a: Total road area A: The land area |
| | Street accessibility [31–33] | The number of times each street segment x is traversed by the "shortest" path between any two other street segments y and z within a particular analysis radius | $Betweeness(x) = \sum\limits_{y\epsilon N} \sum\limits_{Z\epsilon Ry} P(z)OD(y,z,x)$ |
| | Block size [34] | The average side length of the blocks | B_s = B_p/4<br>B_s: The average block size, B_p: The perimeter of a block |
| | Road intersection density [35] | The sum of the road intersections divided by the area | R_id = R_i/A<br>R_id: The road intersection density in the area, R_i: Total number of intersections, A: The land area |
| Typology and Development intensity | Building types [36,37] | The most simplified classification of buildings according to floors and plan configurations | Using the Space Matrix technique:<br>$0 < GSI \le 0.2$: point, $0.2 < GSI \le 0.35$: plate, $GSI > 0.35$: enclosed<br>$0 < $ Floor (average) $\le 3$: low-rise,<br>$3 < $ Floor (average) $\le 6$: multi-rise,<br>$6 < $ Floor (average) $\le 11$: small-high rise,<br>Floor (average) $> 11$: high-rise<br>Ex: $0 < GSI \le 0.2$ and $0 < $ Floor (average) $\le 3$: Low-rise point buildings |
| | FAR [38–40] | Floor-to-area ratio | FAR = F/A<br>FAR: Floor-to-area ratio, F: Total floor area, A: The land area |
| | GSI [39,40] | Ground space index | GSI = B/A<br>GSI: Ground space index, B: Total footprint of buildings in the area, A: The land area |
| | OSR [41] | Open space ratio | OSR = (1−GSI)/FAR<br>FAR: Floor-to-area ratio, GSI: Ground space index |
| | Average floor [40] | The average number of floors | FAR/GSI<br>FAR: Floor-to-area ratio, GSI: Ground space index |
| Land use | Function proportion [45] | The proportion of various functions | A, B, C, D, E, F, G, H, I, J, K, and M are the number of different POIs<br>Function ratio = A:B:C:D:E:F:G:H:I:J:K |
| | Facility density [45] | The total number of facilities divided by the area | F_d = P/A<br>F_d: Facility density, P: The sum of POIs in an area, A: The land area |
| | Average mixed-use index (MXI) [42–44] | The entropy of the various functional POIs divided by the area | $S(p_1, p_2, \ldots, p_n) = -K \sum\limits_{i=1}^{n} p_i log_2 p_i$<br>MXI = S/A, A: The land area |

## 4. Analysis

### *4.1. The Common Features of Urban Morphology in Benchmark Cases*

This section illustrates the common characteristics of indicators in successful coastal cases. The following examines the quantitative analysis results to discuss and interpret its meaning and common features.

#### 4.1.1. Street Network Configuration

Road density and road area density reflect the abundance of roads in an area; the higher the value, the more roads and coverage of roads in the area. Additionally, accessible roads can service various areas and pave the way for developing urban vibrancy [29,30].

There are complete and compact street networks near the shorelines and water and green resources to form the urban areas. Therefore, the study areas have good road density $(14–33.5/km^2)$ and road area density (7–21%). The compact street network enables the blocks with water and green resources to have good road density and road area density. The blocks in or near the regional centers and compact roads have higher road density and road area density values. This is shown in Figures 3 and 4.

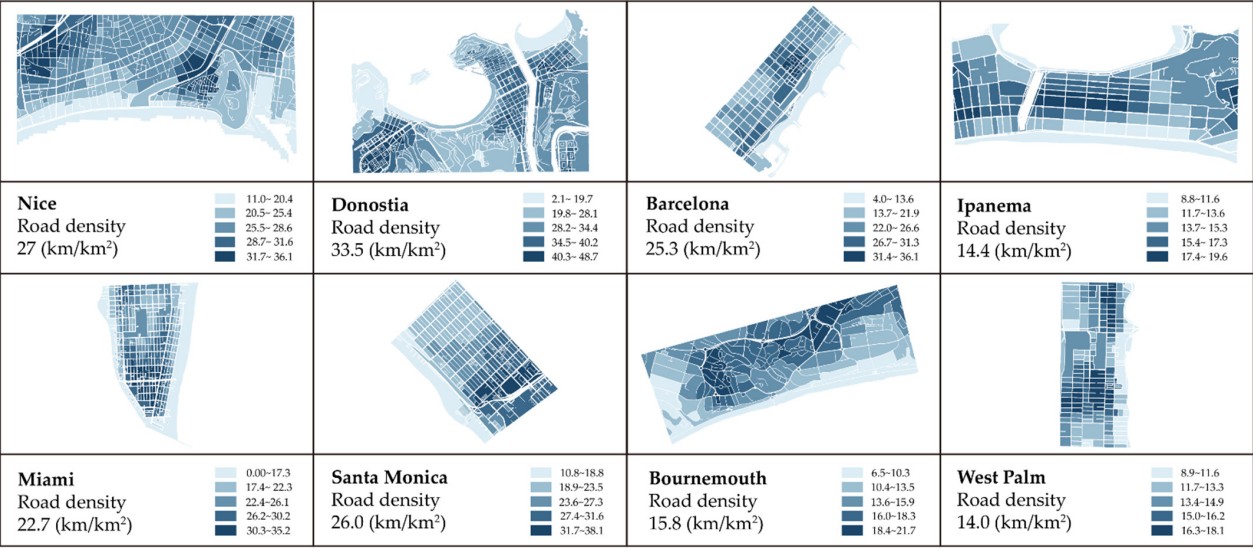

**Figure 3.** The visualization results of road density in successful coastal cases.

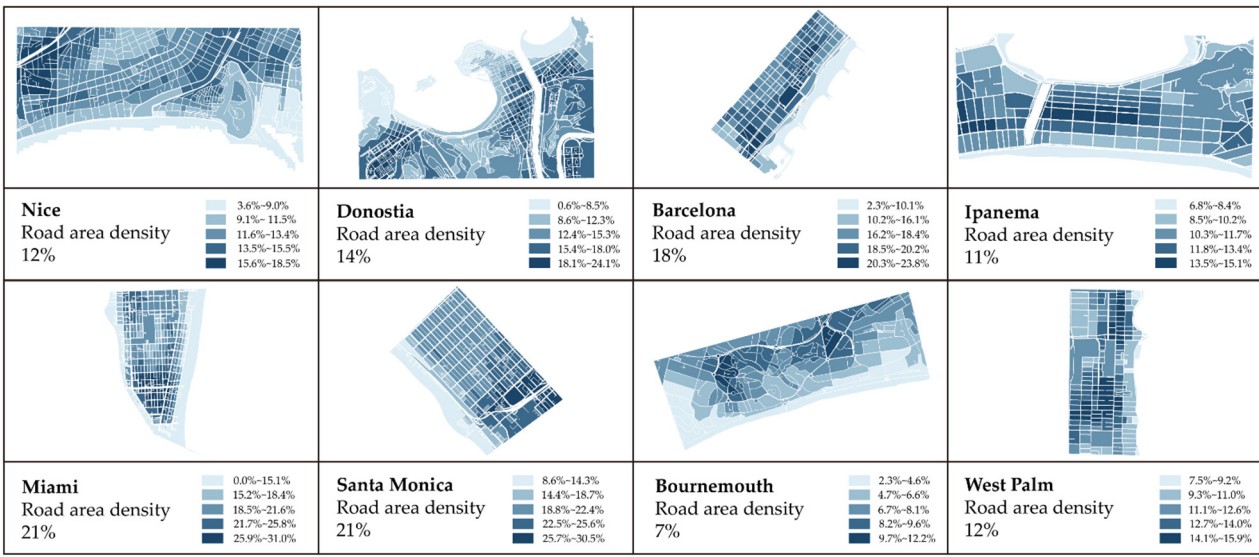

**Figure 4.** The visualization results of road area density in successful coastal cases.

The betweenness of a segment can be obtained using sDNA [32,33], which represents the selection level of the road segment [31]. We utilized BTA400, which means that the measurement is calculated on a pedestrian scale, 400 m, an acceptable 5-min walking distance in the United States [48], and used BTA 5600, which means that the measurement is calculated on a driving scale, 5600 m, that is above the driving scale of 5000 m and is a multiple of 400 m. Generally, areas with high pedestrian choice become regional service centers more easily, and areas with high vehicle choice become city centers more easily [48]. The degree of choice is positively related to urban vibrancy, and streets with high betweenness usually have high urban vibrancy.

- BTA400:

In the benchmark cases, in Figure 5, because of the compact street networks, the study areas have good average values of BTA400, 116.9–1424.9, and there are some segments with high pedestrian betweenness in the study areas. Some are parallel or perpendicular and close to the coastline, and some in the depth of the hinterlands are parallel or perpendicular to the coastline. Segments with high pedestrian betweenness serve as regional service centers [48].

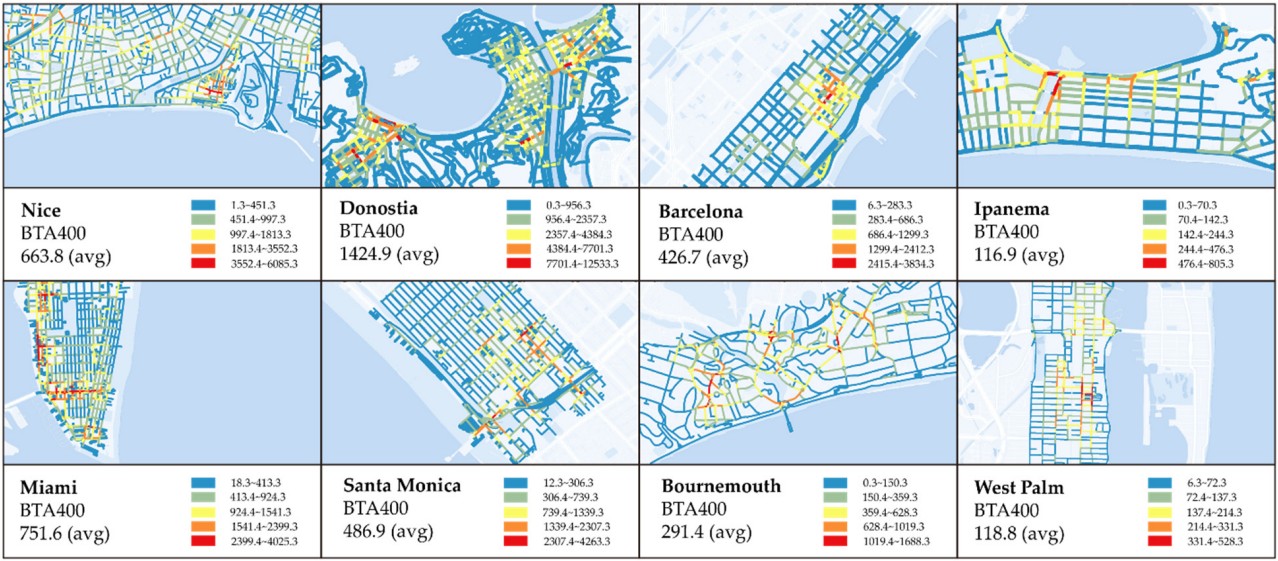

**Figure 5.** The analytical results of BTA 400 in successful coastal cases.

- BTA5600:

As shown in Figure 6, the study areas have good average values of BTA5600, between 6269 and 100,762.2. These successful cases reveal that there are some roads with high betweenness for vehicles in the study areas. Some of them are parallel or perpendicular and close to the coastline, and some are in the depth of the hinterlands and perpendicular or parallel to the coastline. These roads are also usually wider for vehicles and the parts of the city centers.

Road intersection density and average block size can reflect the sizes of the blocks and the ability to turn [34,35]. It is easier to boost urban vibrancy when people can turn and change routes conveniently. Generally, smaller blocks can improve urban vibrancy. Nevertheless, they are not favorable for cars, demonstrating that suitable street blocks are beneficial to urban vibrancy.

Near the shorelines, there are complete, short, and small blocks forming the urban areas. Therefore, the whole areas have a high road intersection density (86.3–533.4/km$^2$) and low block sizes (73.98–85.8 m). The blocks with water and green resources still have a good road intersection density because of the compact block organization. The value of road intersection density gradually increases toward the hinterland until it reaches the highest level in the regional center, as shown in Figure 7.

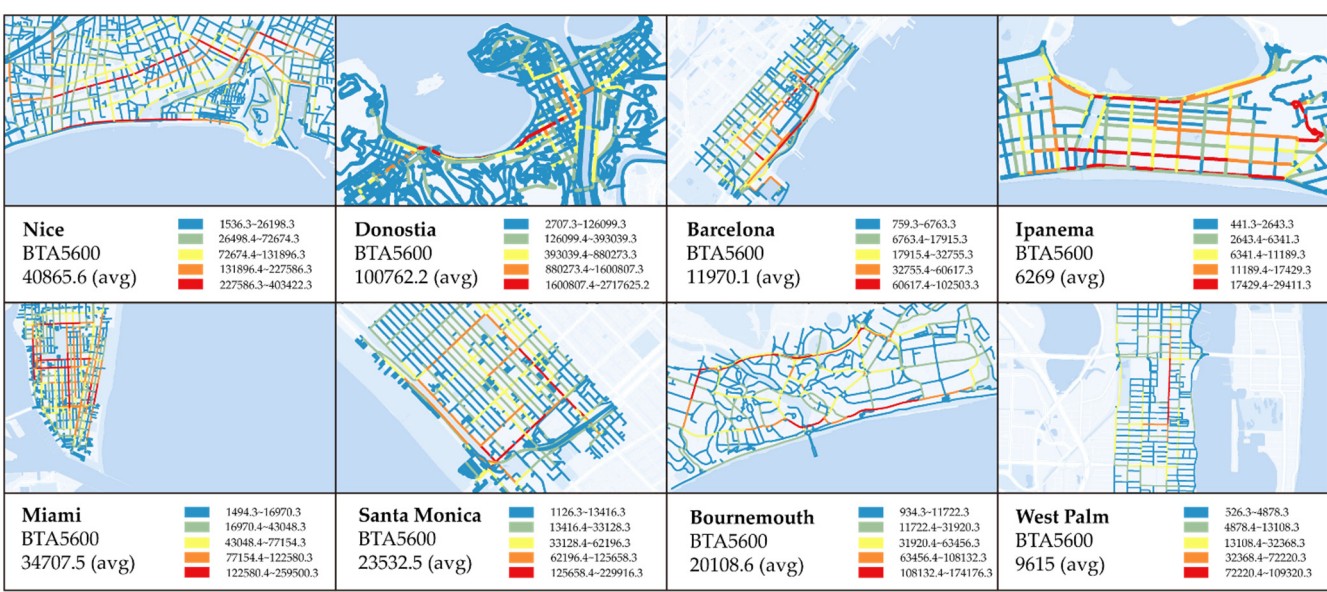

**Figure 6.** The analytical results of BTA 5600 in successful coastal cases.

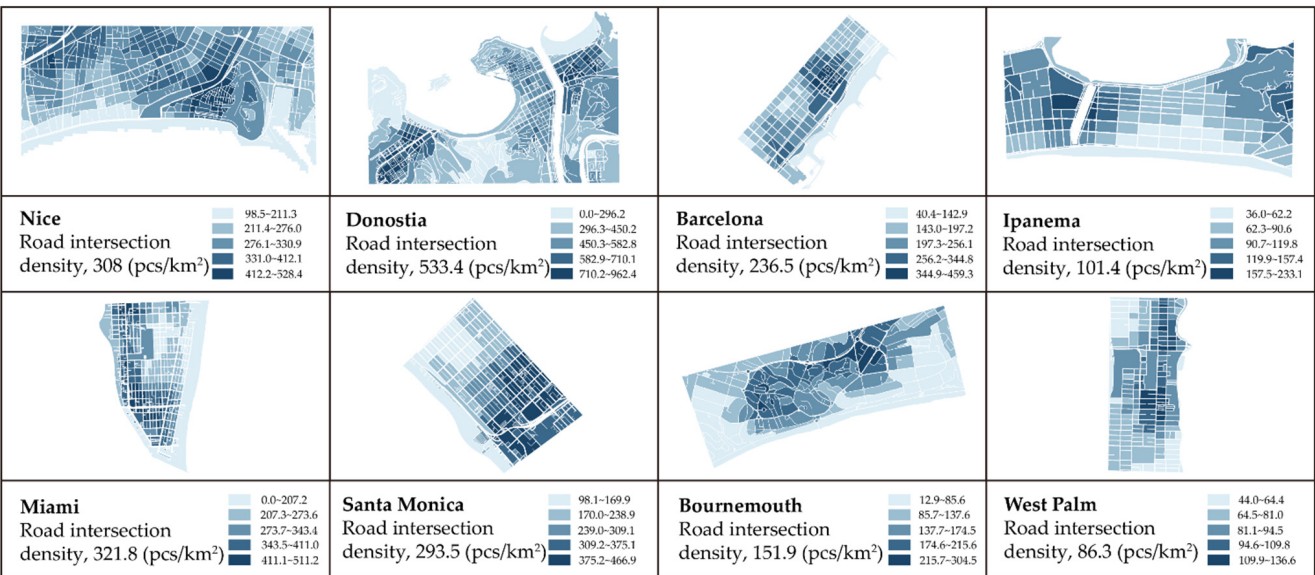

**Figure 7.** The visualization results of road intersection density in successful coastal cases.

The blocks of beaches, mountains, and green spaces form large block sizes, and other compact and short blocks form a completely urban area near these large-sized blocks. These short blocks have square, rectangular, and irregular shapes, as envisaged in Figure 8.

### 4.1.2. Typology and Development Intensity

The spatial matrix method can be used to quickly learn the dominant building types and their composition methods in each block in the coastal area [36,37]. According to the previous understanding of building types and urban vibrancy [24], as highlighted in Table 2, building types can be categorized into low, medium, and high vibrancy forms, depending on their level of urban vibrancy.

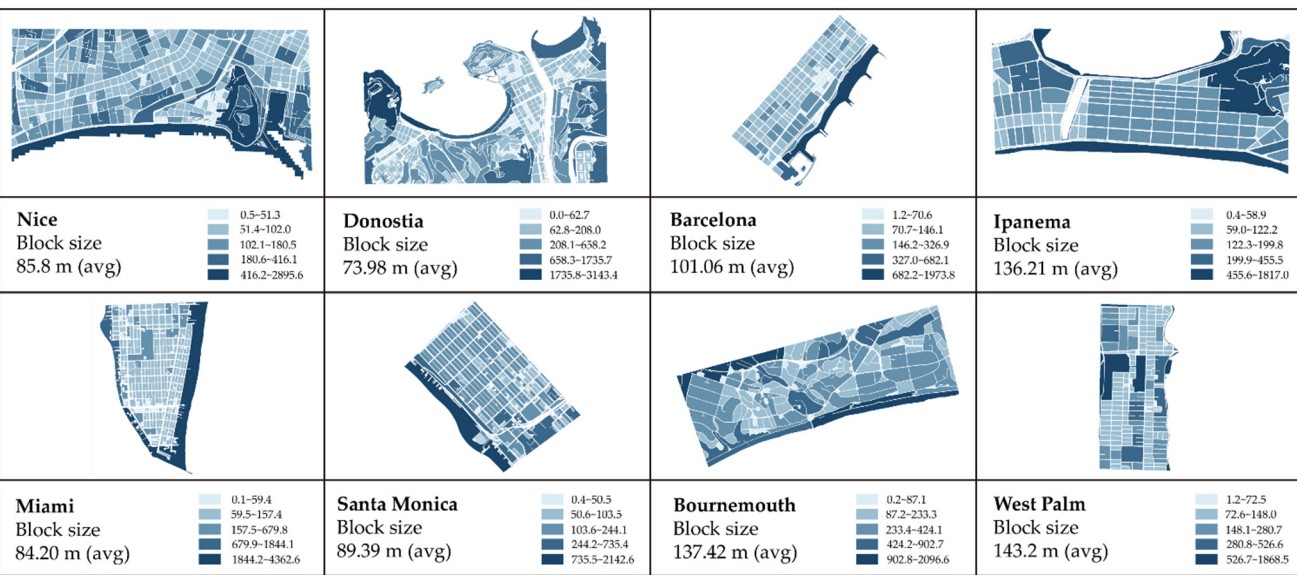

**Figure 8.** The visualization results of average block size in successful coastal cases.

**Table 2.** The urban vibrancy levels of different building types [24].

| Space Matrix | High urban vibrancy | Multi-rise plate buildings, multi-rise enclosed buildings, and (small) high-rise enclosed buildings |
| --- | --- | --- |
| | Medium urban vibrancy | Multi-rise-point buildings, (small) high-rise-point buildings, and (small) high-rise plate buildings |
| | Low urban vibrancy | Low-rise-point buildings, low-rise plate buildings, and low-rise enclosed buildings |

Figure 9 provides the number of blocks dominated by various building types. Consequently, the dominant building types in the whole district can be known.

The successful coastal cases can be categorized into the following forms according to the dominant building types in the regions in Figure 9. These are dominated by:

- Multi-rise and small-high-rise enclosure buildings (vibrancy: high): Nice and Donostia.
- Multi-rise enclosed buildings (vibrancy: high): Barcelona.
- Small-high-rise row houses (vibrancy: high): Ipanema.
- Low-rise and multi-rise row houses (vibrancy: high, low): Miami and Santa Monica.
- Low-rise-point buildings (vibrancy: low): Bournemouth and West Palm Beach.

The building types of cases can be categorized into three forms. The first form is dominated by building types conducive to forming urban vibrancy, such as multi-rise and small-high-rise enclosed buildings and small-high-rise row houses. In the second form, the multi-rise townhouses conducive to urban vibrancy are distributed near the coastline and support the public spaces, while the low-rise townhouses that are not conducive to urban vibrancy are distributed in the hinterland. The third form is dominated in the whole area by the low-rise-point buildings that are not conducive to the vibrancy of the city, but the regional centers have many high-rise row houses that are beneficial to the urban vibrancy to support the regional vibrancy. In Figure-Ground [49,50] (Figure 10), the compositions of the building masses are mainly open blocks. The townhouses or enclosed buildings in the blocks, side by side, form continuous street interfaces, which are advantageous for the formation of urban vibrancy.

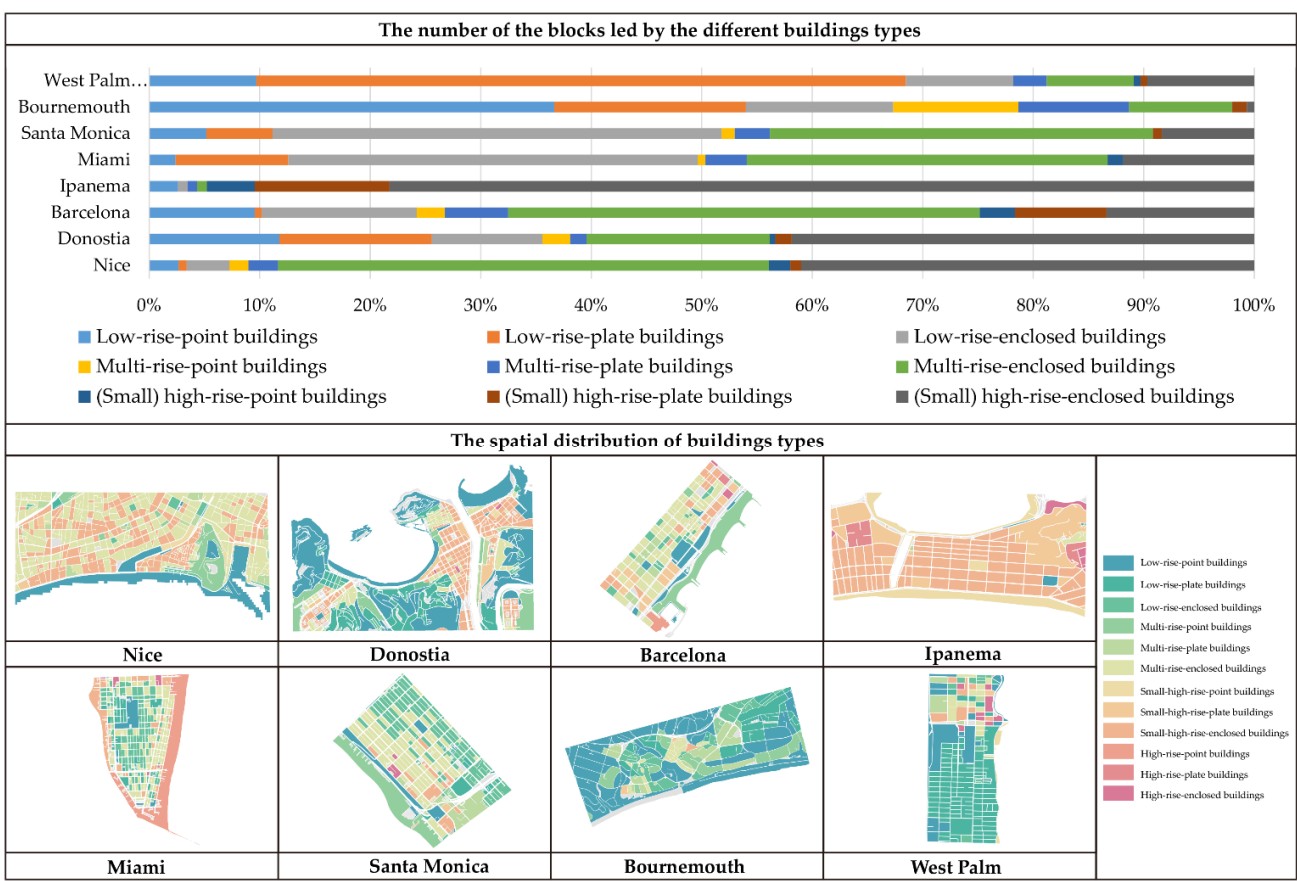

**Figure 9.** The number of blocks led by the different building types and the distribution of building types.

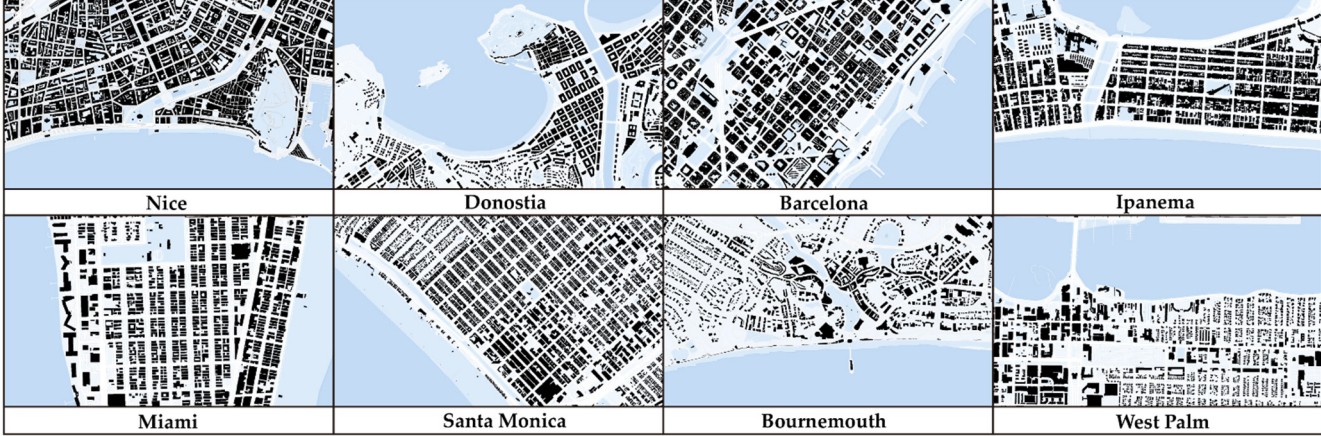

**Figure 10.** The Figure-Ground in successful coastal cases.

The FAR reflects development intensity, and areas with a higher FAR can usually support better urban vibrancy [38–40]. Nevertheless, a higher FAR can also result in a decline in the quality of life; thus, an appropriate FAR range is required. The GSI can be used to determine the development intensity of a site and reflects the coverage of the building area of a region [39,40]. An excessively high coverage will adversely affect fire protection, sanitation, and greening, whereas an excessively low coverage will be unfavorable for urban vibrancy. The open space ratio (OSR) index represents the ratio of vacant land, which means lands without coverage to floor area [41]. The higher the index, the more vacant land available per capita. However, a high index suggests the availability

of more vacant land, which is not conducive to the formation of urban vibrancy. Average floors can reflect the height of the houses in the blocks and affect the development intensity to a certain extent.

　　Because of the compact urban areas in the study areas, they have high development density, with high FAR (63%–310%), high GSI (19-43%), high average floors (3.3–8.2), and low OSR (0.2–1.26). In these cases, the distribution of the blocks with high development intensity can be categorized into three forms, as shown in Figures 11–13. The first form is that the blocks with high development intensity (with high FAR, GSI, and average floors) are all over and form the urban areas near the coastline. The second form is that the blocks with high development intensity (with high FAR, GSI, and average floors) are mainly close to the shoreline to support the services near these huge public spaces, while the development intensity (FAR, GSI, and average floors) of the blocks in the hinterland are low. The third form is that the blocks with high development density (with high FAR, GSI, and average floors) are concentrated in the centers of the areas that are near the coastline with a walkable distance, while the development intensity of the blocks in the remaining areas is low.

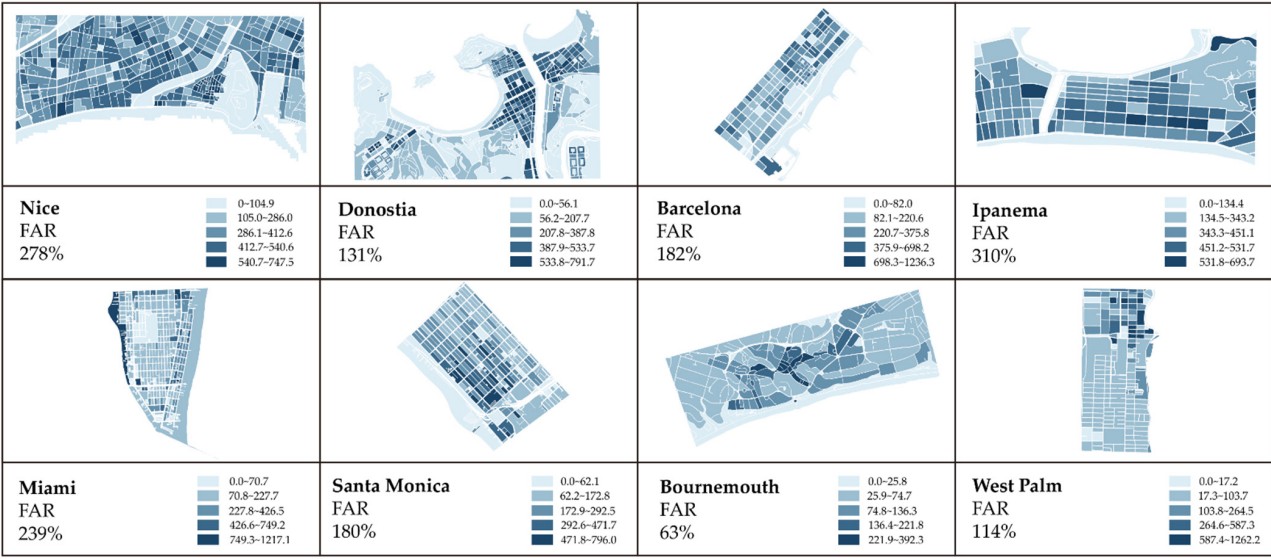

**Figure 11.** The distribution of FAR in successful coastal cases.

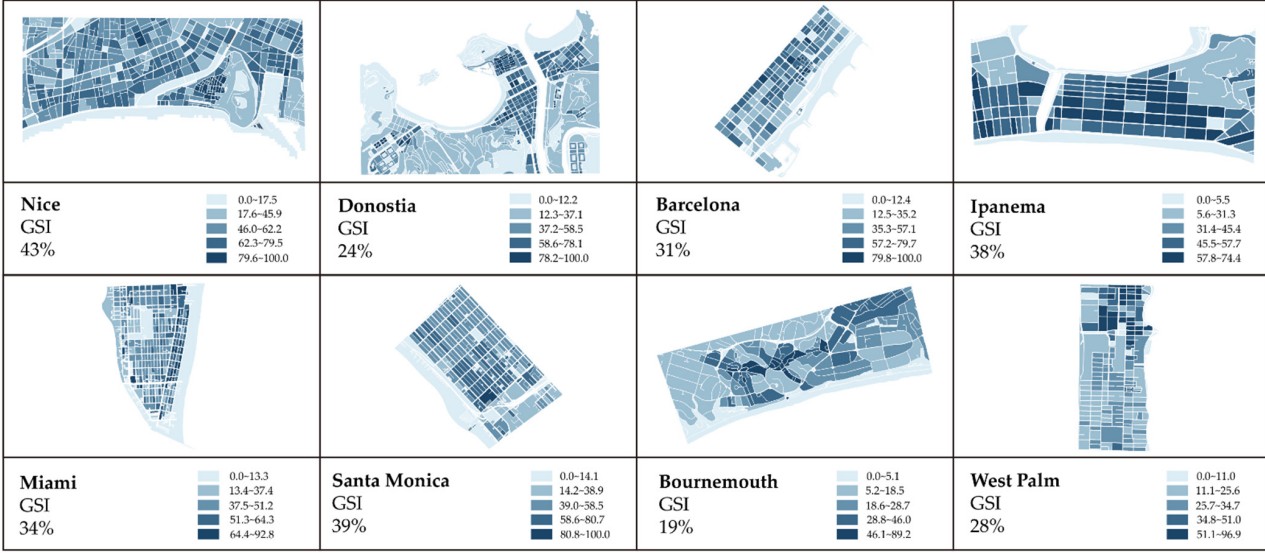

**Figure 12.** The distribution of GSI in successful coastal cases.

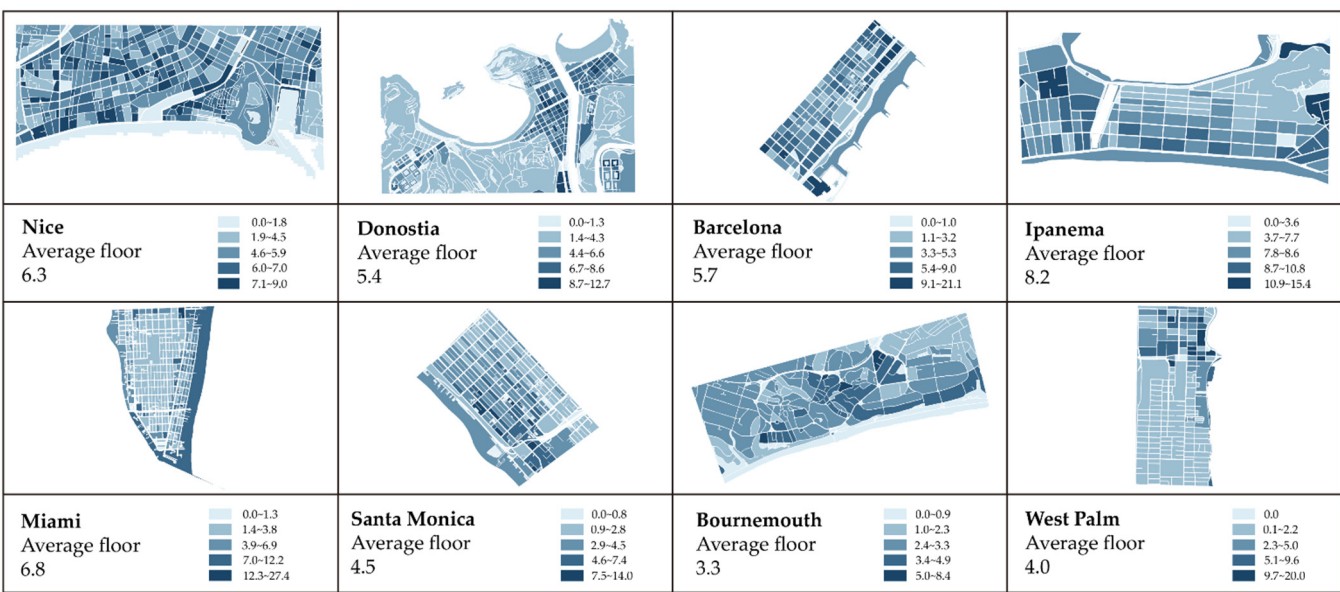

**Figure 13.** The distribution of average floors in successful coastal cases.

In Figure 14, the blocks of coastal beaches, green spaces, and other resources have the highest OSR. However, the blocks near the coastline and in the hinterland are not high. This reveals that the blocks of the urban areas near the coastal resources are all compactly developed. There is a marked contrast between the low OSR of the blocks near the coastline and natural resources and the high OSR of the blocks with natural resources.

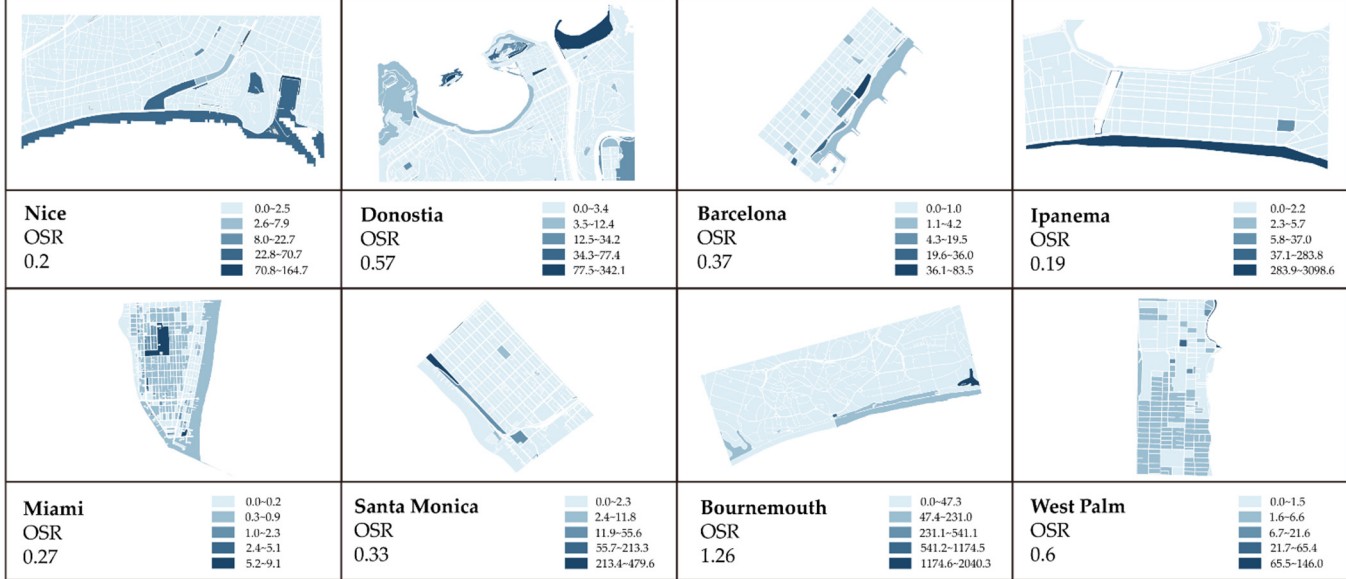

**Figure 14.** The distribution of OSR in successful coastal cases.

4.1.3. Land Use

The facility density reflects the number of facilities in an area [45]. Generally, the higher the facility density, the more convenient and favorable it is for urban vibrancy. The function proportion illustrates the relationship between the various facility functions and is an important factor affecting vibrancy [45]. The MXI reflects the functional mixing situation of the region [42–44]. Generally, the higher the MXI, the more favorable it is to urban vibrancy.

The study areas have high land use with a high facility density (416.4–2687.4/km²) and a high MXI (30.6–127.9/km²). As shown in Figures 15 and 16, the land use of the cases can be categorized into three forms. The first form is that the blocks with high land use (with a high MXI and facility density) are all over and form an urban area near the coastline. The second form is that the blocks with high land use (with a high MXI and facility density) are mainly close to the shoreline, while the low land use (with a low MXI and facility density) blocks are in the hinterland. The third form is that the blocks with high land use (with a high MXI and facility density) are concentrated in the center of the area, and the values of the remaining areas are low. Furthermore, there is a high proportion of life, shopping, financial, medical, lodging, and food services in the successful coastal cases (Figure 15).

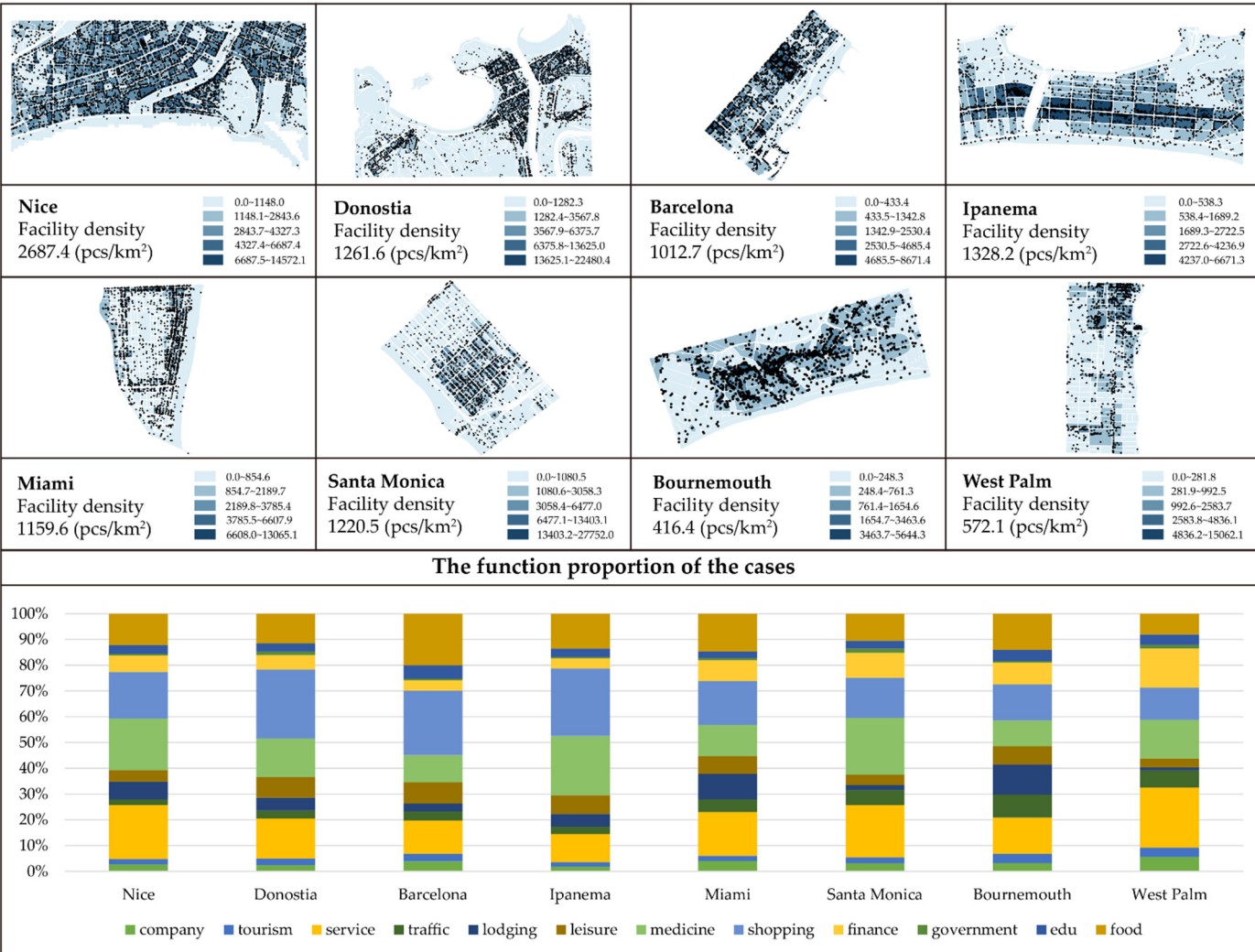

**Figure 15.** The distribution of facility density and the function proportion in successful coastal cases.

4.1.4. The Summary of the Common Range of Each Morphological Indicator in the Benchmark Cases

The quantitative urban form analysis of the benchmark cases could aid in summarizing the common range of morphological indicators of creating a successful waterfront area. Table 3 presents the values of those indicators in different cases, including the minimum, maximum, average values, and standard deviation. To be specific, in the street network configuration aspect, the road density, road area density, road intersection density, and block size are expected to be set between 14 and 33.5, 7 and 21, 86.3 and 533.3, and 73.9 and 143.2, respectively. Second, the FAR, GSI, OSR, and average floor are expected to be

set between 0.63 and 3.1, 19 and 43, 3.3 and 8.2, and 30.6 and 127.9, respectively. Finally, the MXI and facility density are expected to be set between 30.6 and 127.9, and 416.4 and 2687.4, respectively.

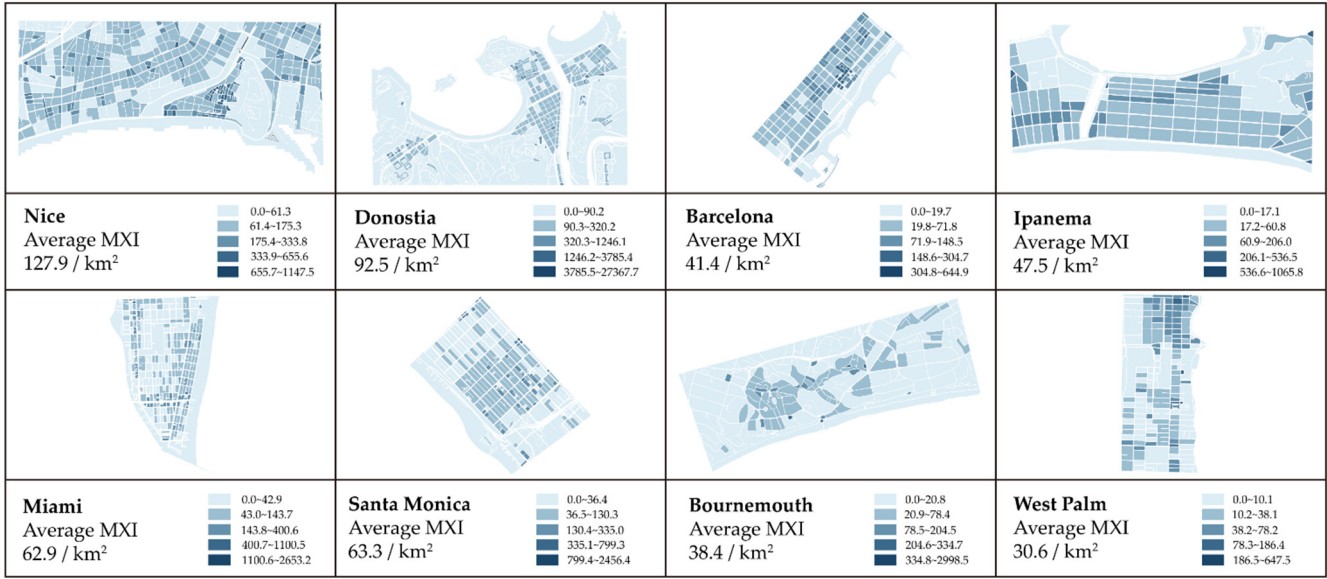

**Figure 16.** The distribution of MXI in the successful coastal cases.

**Table 3.** Quantitative analysis summary.

| | Street Network Configuration | | | | | | Typology and Development Intensity | | | | Land Use | |
|---|---|---|---|---|---|---|---|---|---|---|---|---|
| Indicators | Road density (km/km²) | Road area density (km²/km²) | Road intersection density (pcs/km²) | Block average size (m) | BTA 400 | BTA 5600 | FAR (%) | GSI (%) | OSR (km²/km²) | Average floor | MXI (/km²) | Facility density (pcs/km²) |
| Minimum value | 14.0 | 7.0 | 86.3 | 73.9 | 116.9 | 6259 | 63 | 19.0 | 0.2 | 3.3 | 30.6 | 416.4 |
| Maximum value | 33.5 | 21.0 | 533.4 | 143.2 | 1424.9 | 100,762.2 | 310 | 43.0 | 1.3 | 8.2 | 127.9 | 2687.4 |
| Average value | 22.3 | 14.5 | 254.1 | 106.4 | 535.1 | 30,978.8 | 183 | 32.0 | 0.5 | 5.5 | 63.1 | 1207.3 |
| Standard Deviation | 6.5 | 4.7 | 136.60 | 26.2 | 399.5 | 28,686.4 | 78.4 | 7.5 | 0.3 | 1.4 | 30.5 | 640.0 |

Through the comparison of those values between the benchmark and problematic cases, precise guidance for the problematic area with low urban vibrancy can be obtained. That is demonstrated in detail in the next section.

### 4.2. Design Applications of Theoretical Insights: The Suggestions for the Jinshatan Area as an Example

Jinshatan is a waterfront tourism resort located in Yantai. Built over the last two decades, it comprises several tourist and leisure facilities along the waterfront area. Compared to the benchmark cases that have a long history, Jinshatan lacks urban vibrancy. As shown in Figure 17, this research explores the reason for this gap between Jinshatan and the benchmark cases through the analysis of the urban morphological indicators. The quantitative analysis results offer some operational guidance on the urban form.

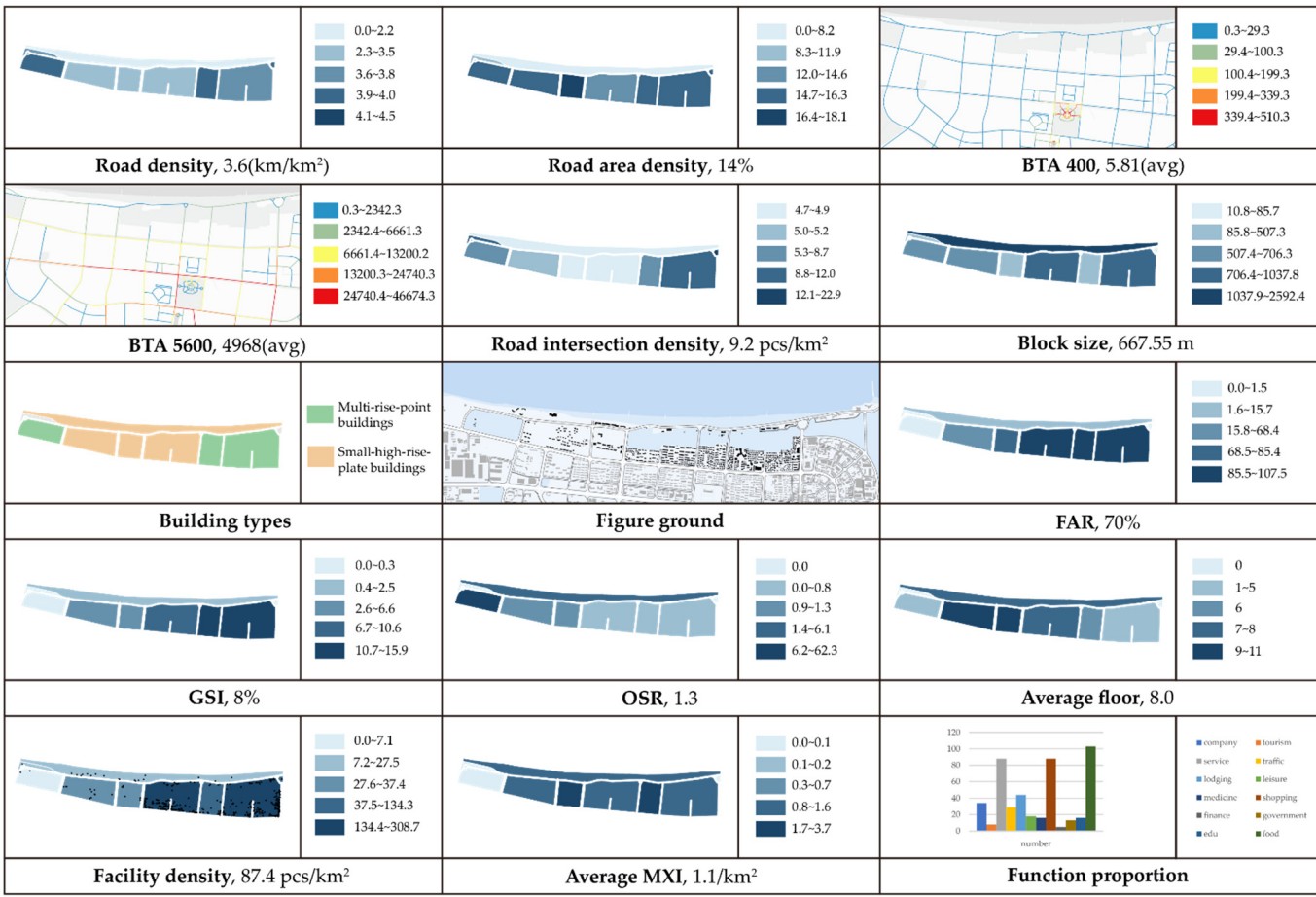

**Figure 17.** The analysis result of the Jinshatan area, Yantai.

- Suggestions for Jinshatan's street network configuration

As shown in Figure 17, Jinshatan has a morphological structure with a sparse street network and huge blocks. It would be ideal to reform it into an urban area with a compact street network and shorter blocks to increase its accessibility and develop some regional or city centers. Specifically, through the comparison between Jinshatan and the successful cases highlighted in Figure 18, the road density could be changed from 3.6 to 14–33.5, and the road area density could be changed from 14% to 21%. Additionally, road intersection density could be increased from 9.2 to 86.3–533.4, and the average block size may be reduced from 667.5 to 73.9–143.2. It is recommended that the BTA400 be promoted from 5.1 to 116.9–1424 and the BTA5600 from 4968 to 6259–100,762.2.

- Suggestions for the typology and the development intensity of Jinshatan, Yantai

As shown in Figure 17, Jinshatan features multi-rise-point buildings and small-high-rise plate buildings that contribute to medium urban vibrancy. The analysis recommends the addition of building types that are conducive to urban vibrancy, including multi-rise and small-high-rise enclosed buildings and multi-rise and small-high-rise row houses. Additionally, these buildings can be arranged at the Jinshatan waterfront evenly or concentrated in the center of the hinterland, near the coastline. Furthermore, the gated communities could also be changed to open blocks to form continuous street interfaces.

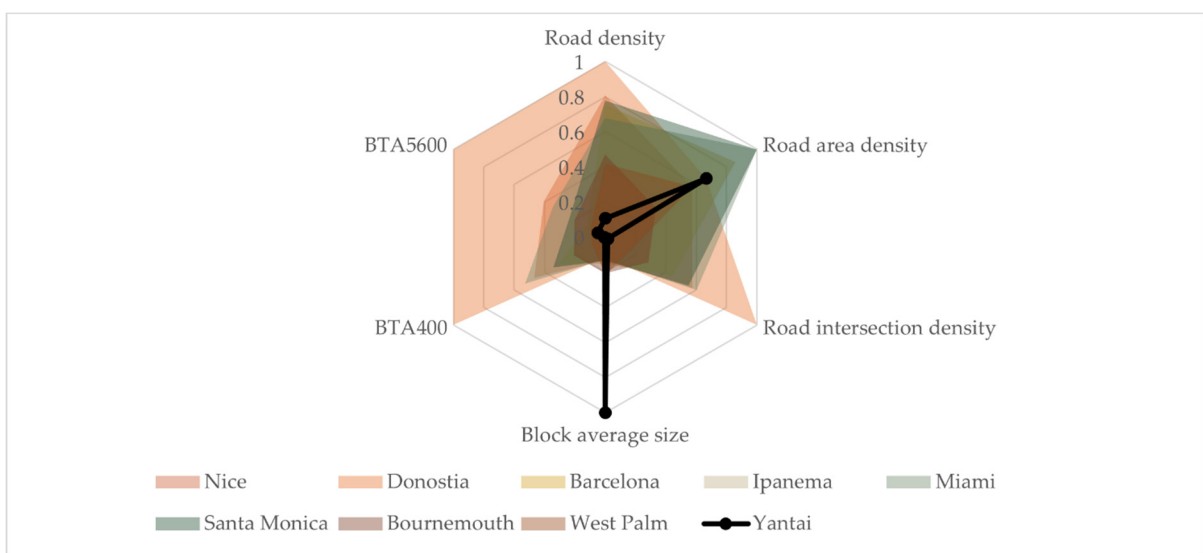

**Figure 18.** The suggestions for the street network configuration.

The development density could be improved to form a compact urban area that can support urban vibrancy. As to the specific guidance of indicator range, in Figure 19, it is recommended that the FAR for Jinshatan be changed from 70 to 63–310, the GSI from 8 to 19–43, and the OSR from 1.3 to 0.19–1.26. The following strategies relating to development intensity are recommended: (1) the development intensity can be increased evenly throughout the waterfront area, (2) the high development intensity blocks should be arranged close to the shoreline and the low development intensity blocks in the hinterland, and (3) the high development intensity blocks should be arranged close to the regional centers, with the remaining area retaining low development intensity.

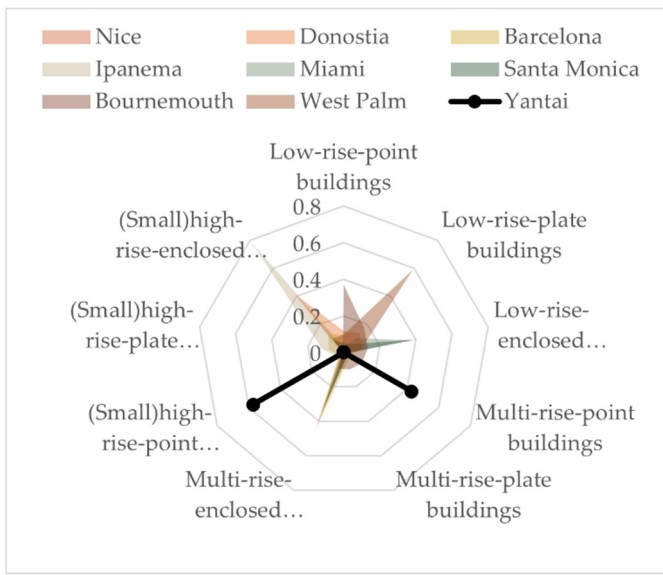
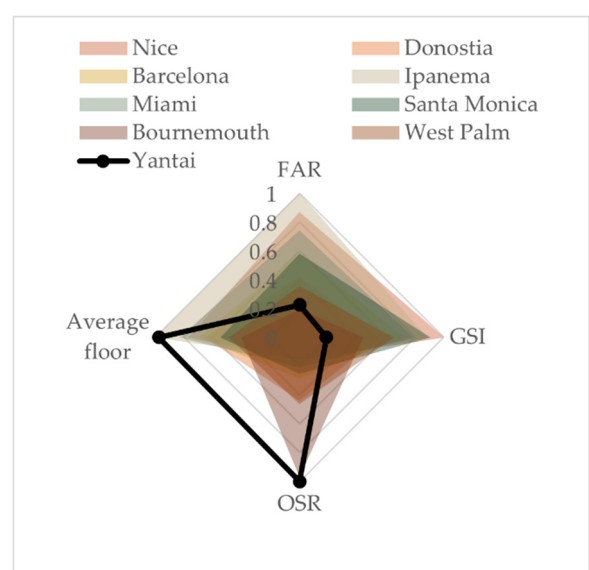

**Figure 19.** The suggestions for the typology and the development intensity.

- Suggestions for Jinshatan's land use

As shown in Figure 17, the Jinshatan waterfront area lacks dense functional facilities, with a low level of function mixture. It is recommended that the overall number of facilities be increased, raising the facilities' density from 87.4 to 416.4–2687.4. The MXI could also be increased from 1.1 to 30.6–127.9. Additionally, as shown in Figure 20, Jinshatan mainly provides dining, living, and shopping services. The proportion of finance and medical

services should be improved. In terms of the spatial distribution of land use, the following strategies could be adopted: 1) high land use blocks in the whole area, 2) arrangement of high land use blocks close to the shoreline, with low land use blocks in the hinterland, and 3) arrangement of high land use blocks near the regional centers, while that of the low land use blocks in the remaining area.

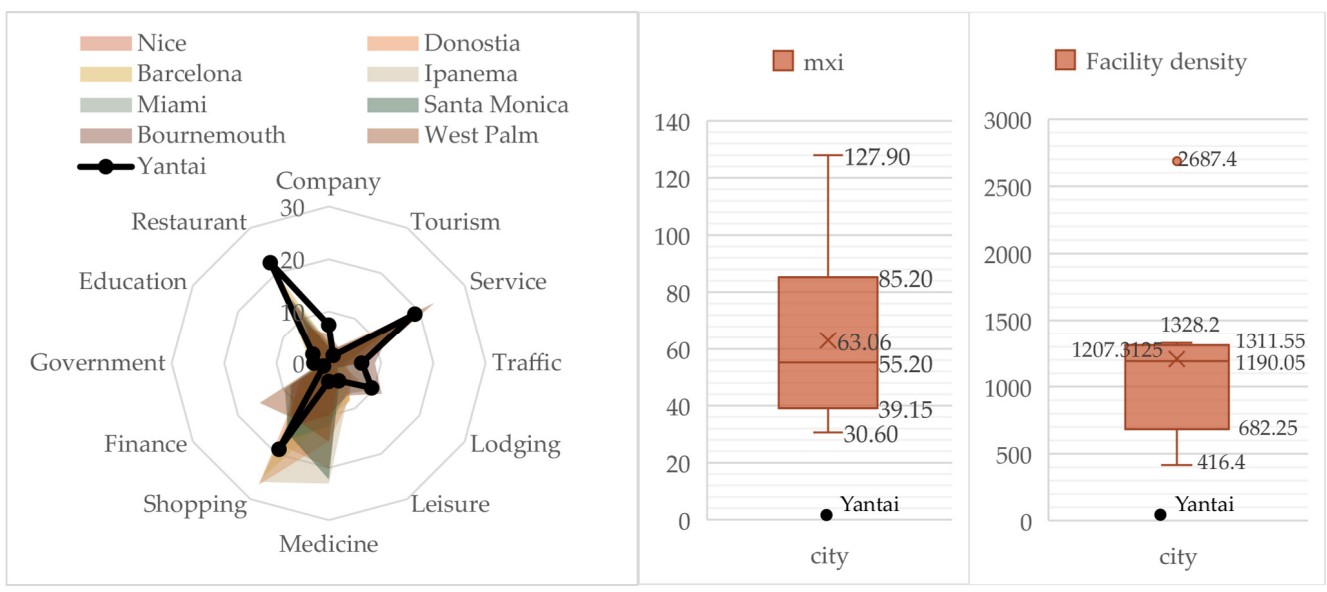

**Figure 20.** The suggestions for land use.

## 5. Discussions and Conclusions

### 5.1. Summary and Contribution of This Research

This study offers twofold contributions. It summarizes the common characteristics of successful waterfronts with a high degree of vibrancy. The features were extracted by the analysis of the morphological indicators related to vibrancy from representative benchmark cases, which were selected from travel websites and under strict requirements [25]. In addition, the comparative analysis of Section 4.2. shows that, despite the differences in the benchmark cases, they have greater differences from the problem case. Therefore, the benchmark cases could reflect the common urban form features to some extent of the vibrant waterfronts. To be specific, it can be found in the illustrations in Sections 4.1 and 4.2, that the vibrant waterfronts have compact street networks and building types that are conducive to public activities. They also possess high development intensity and mixed-dense land use. Thus, compact urban areas are recommended for developing vital waterfronts. These conclusions are consistent with the classical Jacob's urban theory. Most importantly, this study gives a specific range of relevant morphological indicators from the quantitative urban morphology, which will tell us what a reasonable level of indicators should be. For example, based on the results of this study, the road density for a vibrant waterfront is suggested to be set in the range of 14.0–33.5 km/km$^2$ (Figure 3), while the floor area ratio should be set in the range of 0.63–3.1 (Table 3).

This study also provides an operational approach for developing the urban vibrancy of the waterfronts based on the quantitative urban morphological analysis. Comparing the indicators between Jinshatan (the problematic case) and the famous coastal waterfront cases, the following insights were obtained. In terms of the streets, blocks, and development intensity, the average block size and OSR should be reduced in Jinshatan, while the road density, road area density, FAR, and GSI should be increased correspondingly. As for the building types, an appropriate increase is needed in the proportion of small high-rise enclosed buildings, and multi-rise and small high-rise row houses. For land use, there is a large gap in the functional density and mixture between Jinshatan and the other case areas. Both should be enhanced by introducing multiple and complex

functional services. Additionally, the proportion of medicine and leisure facilities could be increased appropriately. Overall, this study provides a comprehensive and operational path to enhance the vibrancy of the coastal waterfront from the perspective of quantitative urban morphology.

*5.2. Research Limitations and Future Perspective*

This study has several limitations. Firstly, for the methods of case selection, due to the difficulty of using quantitative methods, the benchmarks were selected by a qualitative approach based on the predefined conditions (Section 3.1.1). More cases should be included in the future, and quantitative sampling methods should be attempted to bring more robust results. Secondly, human-scale morphological indicators, such as street interfaces, were also identified as important factors influencing the vibrancy of public space. In addition, more features, such as pedestrian roads and the availability of public spaces, may be related to urban vibrancy. We may take them into account in the future. Thirdly, the results of this study also provide some clues to discovering the classification of urban morphology, for which different types of coastal waterfronts may have different paths of vibrancy enhancement. Under the constraints of costs and benefits, their paths need to be further explored. Fourthly, although current study mainly focuses on urban morphological indicators, we would involve the indicators from social dimensions in our future study to achieve a more systematic consideration.

**Author Contributions:** Conceptualization, Y.Y.; methodology, Y.Y. and Y.H.; software, L.S.H.; validation, Y.Y. and Y.H.; formal analysis, L.S.H. and Y.H.; investigation, L.S.H. and Y.H.; resources, Y.H.; data curation, L.S.H. and Y.H.; writing—original draft preparation, L.S.H. and Y.H.; writing—review and editing, Y.Y. and Y.H.; visualization, L.S.H.; supervision, Y.Y.; project administration, Y.Y.; funding acquisition, Y.Y. All authors have read and agreed to the published version of the manuscript.

**Funding:** This research was funded by National Natural Science Foundation of China, grant number 52078343 and 51838002, Natural Science Foundation of Shanghai, grant number 20ZR1462200, and Fundamental Research Funds for the Central Universities, grant number 22120210540.

**Data Availability Statement:** Not applicable.

**Acknowledgments:** We would like to thanks Changyu Chen, Xiaoyu Chen and Chengcheng Huang for their support on data preparation.

**Conflicts of Interest:** The authors declare no conflict of interest.

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
