# Peer review of "Coastal Waterfront Vibrancy: An Exploration from the Perspective of Quantitative Urban Morphology"

_buildings, doi:10.3390/buildings12101585_

Round 1

Reviewer 1 Report

Dear Colleagues,

This is an interesting research with valuable purpose, but I have some comments, suggestions that will improve it. 

Considering the Figures 18, 19 and 20, I could recommend its improvement that will aimed in its readability - to use different colour scheme (different colours), or to try with other type of diagram.

Also, in Figure 20 there is a diagram that have local symbols and signs, which need to be presented, translated in English.

Best regards

Author Response

Response to reviewers

First, we highly appreciate the editor and the two anonymous reviewers taking the time to offer us comments and insights related to the paper. We greatly appreciate the opportunity to extend our thinking and improve our manuscript. The responses are given according to the sequence of the reviewer’s comments. All the revised parts in our manuscript are marked in blue for your kind reference.

-------------

Reviewer #1: This is an interesting research with a valuable purpose, but I have some comments, and suggestions that will improve it.

Comment 1: Considering Figures 18, 19, and 20, I could recommend its improvement that will aimed in its readability - to use different color scheme (different colors), or to try with other type of diagrams.

Also, in Figure 20 there is a diagram that has local symbols and signs, which need to be presented and translated into English.

Response:

Many thanks for your kind help. We have changed the color gradient and transparency of Figures 18, 19, and 20 to enhance their readability. We also revised the local symbols in Figure 20.

Again, we appreciate all your insightful comments. We would like to express our appreciation to the editor and anonymous reviewers in the acknowledgment for the time and effort you have taken to provide such insightful guidance. We would be glad to respond to any further questions and comments that you may have.

Reviewer 2 Report

interesting topics that needs to be better investigated. the methodology is weak in many steps and the selection of benchmarks does not seem to be ''robust'' even in qualitative terms. please see notes and requests for clarification in the attached text

Author Response

Response to reviewers

First, we highly appreciate the editor and the two anonymous reviewers taking the time to offer us comments and insights related to the paper. We greatly appreciate the opportunity to extend our thinking and improve our manuscript. The responses are given according to the sequence of the reviewer’s comments. All the revised parts in our manuscript are marked in blue for your kind reference.

-------------

Reviewer #2: interesting topics that need to be better investigated. the methodology is weak in many steps and the selection of benchmarks does not seem to be ''robust'' even in qualitative terms. please see notes and requests for clarification in the attached text.

Comment 1: Why not urban vibrancy? there's a lot of literature on it

Response:

Thanks for your comments. We have changed the term “vitality” to “vibrancy” in the whole manuscript.

Comment 2: Historical values and economic development advantages seem to be in contrast... Please explain it better.

Response:

Thanks for your help. This sentence has been reorganized to make it clear. The detailed revisions have been attached below.

“Thus, they tend to have important historical value and potential for tourism development, benefiting economic development to some extent.” (Section 1.1, Page 1, Line 33-35)

Comment 3: better clarify the sentence: do you mean that they are not urban waterfronts (so they and are therefore more naturalistic), or that, even though they are in urban areas, they should not be vibrant?

Response:

Thanks for your helpful advice. The sentence has been clarified. Detailed revisions have been attached below.

“Consequently, urban vitality is a key indicator of spatial quality, related to the livability, attractiveness, and sustainability of waterfronts.” (Section 1.2, Page 2, Line 53-55)

Comment 4: missing references, or case studies…

Response:

Thanks for your helpful advice. The references have been added. Detailed revisions have been attached below.

“Although many studies have proved the correlation between urban morphological characteristics and urban vibrancy [5,6], it still lacks a systematic understanding of the common urban form features of the vibrant waterfronts.” (Section 1.3, Page 2, Line 58-61)

Comment 5: Which one?

Response:

We have added the key features below.

“In general, there is a broad consensus on the spatial characteristics that are conducive to vitality such as street intersection density, building density, block size and land use diversity.” (Section 2.1, Page 2, Line 92-94)

Comment 6: Please try to outline the considered aspect for each cited reference, in a final table. please add to the previously requested table, the appropriate indicators found in the literature.

Response:

Thanks for pointing it out. In Table 1, we have listed the aspects and indicators considered and cited references.

Comment 7: Do you mean a range of value (i.e. from 0 to +1) or properly a set of indicators? or dimensions?

Response:

Thanks for your question. The sentence has been revised to make it clearer. A detailed revision has been attached below.

“Second, although the correlation coefficients between key morphological features and urban vibrancy have been analyzed, detailed thresholds of these key elements of urban form are often missed. An understanding of the valid range of thresholds would help to provide precise guidance on urban design practice..” (Section 2.3, Page 3, Line 120-124)

Comment 8: missing definitions!

Response:

Many thanks for your help. As to the definitions of “common intervals”, we acknowledge that this terminology is inappropriate. Thus, it has been amended to “the range of values for each morphological indicator”.

“First, the successful benchmark waterfronts in different coastal cities are selected, and the common urban form characteristics of those benchmarks are summarized.” (Section 2.3, Page 3, Line 126-128)

“Subsequently, we summarized the overall range of values for each morphological indicator of those benchmark cases.” (Section 3., Page 3, Line 136-137)

Comment 9: why? please be explicit with reference to the literature review...

Response:

Thanks a lot for your help. The literatures supporting the dimensions of morphological indicators can be found in section 3.2. Thus, the reference has been added at the beginning to make it clear. The detailed revisions have been attached below.

“Secondly, we identified the urban morphological indicators in three dimensions, including street network configuration, land use, typology, and development intensity. Detailed sources for the three dimensions can be found in section 3.2.” (Section 3, Page 3, Line 132-136)

“Starting from the “Anglo-Germanic Historico-geographers” and “town-plan analysis” research methods proposed by Conzen [20], the urban form can be disassembled into town planning, typology models, and land use models. Town planning can be divided into street organizations, plots, blocks, and buildings. According to the definitions mentioned above, to summarize the relevant features, the quantitative urban morphology can be divided into three dimensions as shown in Table 1, including street network configuration, typology and development intensity, and land use. The meaning of each indicator has also been explained below.” (Section 3.2, Page 5, Line 174-181)

“…… according to the classical theory of urban vibrancy and urban form [5,6,26-28], the indicators affecting urban vibrancy can be extracted from each dimension. Specifically, road density, road area density, street accessibility, block size, and road intersection density are extracted from street network configuration. Generally, compact road networks, short blocks, and high-accessibility areas contribute to urban vibrancy [29-35]. From the dimension of typology and development intensity, building types, floor area ratio, ground space index, open space ratio, and average building floors can be extracted. Usually, areas with high development intensity may contribute to urban vibrancy [36-41]. Furthermore, several kinds of building types are conducive to urban vibrancy. In addition, the average mixed-use index, the proportion of functions, and the density of facilities are extracted from the land use dimension. Empirical studies show that the higher degree of mixing and density of facilities in an area, the more conducive it is to the vibrancy of the city [42-44]. The function proportion also affects urban vibrancy [45,46]. Table 1 lists urban morphological types, indicators, definitions, and formulas in detail.” (Section 3.1.2., Page 5, Line 182-196)

Comment 10: all over the world? do you think that the location (in the US or in Europe or in Asia) could be significant and could change the selected dimensions/ indicators?

Response:

Many thanks for pointing out this question. Firstly, we acknowledge that the indicators related to vitality may vary depending on the environmental context. Thus, the selected cases and urban morphology dimensions for this study were mainly from the US and Europe (Please refer to sections 3.1 and 3.2). Secondly, we agree that the selection of the cases may strongly affect the common ranges of these indicators. Consequently, we had to carefully select cases according to the conditions as illustrated in section 3.1. Detailed revisions have been attached below:

“Accordingly, through the world's leading travel website, tripadvisor.com, we found some famous vibrant waterfronts located in the USA and Europe. Then we selected the study areas conforming to the conditions above as benchmark cases [25]. The study area of each case is from the coastline to the hinterland one kilometer away (a ten-minute walking distance), with an area of 3 to 5 square kilometers.” (Section 3.1.1., Page 4, Line 150-154)

Finally, it is also necessary to acknowledge our limitations in case selection in the text. Detailed revisions have been attached below.

“Firstly, for the methods of case selection, due to the difficulty of using quantitative methods, the benchmarks were selected by a qualitative approach based on the pre-defined conditions (section 3.1.1). More cases should be included in the future, and quantitative sampling methods should be attempted to bring more robust results.” (Section 5.2, Page 21, Line 476-480)

Reference:

TripAdvisor. Available online: https://www.tripadvisor.com/TravelersChoice (accessed on 12 September 2022)

Comment 11: missing source. on what basis these 3 indicators were chosen? how can they be measured?

Response:

Thanks for your question. For the literature support for the 3 dimensions of morphological indicators, please refer to section 3.2. The detailed text has also been attached below.

“Starting from the “Anglo-Germanic Historico-geographers” and “town-plan analysis” research methods proposed by Conzen [20], the urban form can be disassembled into town planning, typology models, and land use models. Town planning can be divided into street organizations, plots, blocks, and buildings. According to the definitions mentioned above, to summarize the relevant features, the quantitative urban morphology can be divided into three dimensions as shown in Table 1, including street network configuration, typology and development intensity, and land use. The meaning of each indicator has also been explained below.” (Section 3.2., Page 5, Line 174-181)

Reference:

Conzen, M.R.G. Alnwick, Northumberland: A Study in Town-Plan Analysis. Transactions and Papers (Institute of British 500 Geographers) 1960, iii-122, doi:10.2307/621094.

Reference:

Conzen, M.R.G. Alnwick, Northumberland: A Study in Town-Plan Analysis. Transactions and Papers (Institute of British 500 Geographers) 1960, iii-122, doi:10.2307/621094.

Comment 12: What is the output? a table? a database? a GIS?

Response:

Thanks for your question. The output of the summary of common morphological characteristics among those benchmarks includes table 3 in section 4.1 and figures 18-20 in section 4.2.

Comment 13:

Maybe you should have a set of different benchmarks to do this comparison…

Response:

Thanks for this helpful comment. We will include more cases in the future and this deficiency has been added as a reflection of this study in section 5.2.

“More cases should be included in the future, and quantitative sampling methods should be attempted to bring more robust results.” (Section 5.2, Page 21, Line 478-480)

Comment 14: Do you really think they can be compared???

Response:

We acknowledge that it is difficult to find a quantitative way of measuring the vibrancy of benchmark cases to verify the comparability of those benchmark cases. Thus, the comparability of those benchmark cases has been explained in two aspects. Firstly, the commonalities of those benchmarks are introduced in section 3.1.1. Second, the comparative analysis of 4.2 shows that despite the differences in the benchmark cases, they have greater differences from the problem case. Therefore, they are relatively close in urban form. Detailed explanations have been added in section3.1.1. and 5.1.

 “Although the cases are from different places and even have different morphological characteristics in some aspects, they are all famous for their urban vibrancy and meet the following environmental conditions. Consequently, we assume that the famous waterfront urban areas might have common ranges of indicators in urban morphology. Their common ranges might be able to form advice for the Jinshatan area, a waterfront area with a mixed function but lacking urban vibrancy.” (Section 3.1.1., Page 4, Line 156-162)

“Besides, the comparative analysis of 4.2 shows that despite the differences in the bench-mark cases, they have greater differences from the problem case. Therefore, the benchmark cases could reflect the common urban form features to some extent of the vibrant waterfronts.” (Section 5.1, Page 20, Line 448-451)

Comment 15: Please support the statement or refer it to appropriate sources.

Response:

Many thanks for this advice. the text has been revised to make its meaning clearer. The relevant sources have been added. The detailed revision has been attached below.

“The Jinshatan area has numerous geographical and environmental advantages. However, based on the relevant reviews on the website of Tripadvisor.com [25], it can be found that compared to other famous waterfronts area in china, this area has a lower level of tourism prosperity.” (Section 3.1.2, Page 5, Line 169-172)

Reference:

TripAdvisor. Available online: https://www.tripadvisor.com/TravelersChoice (accessed on 12 September 2022)

Comment 16: it seems to be replicated... length and area, but always streets, divided by the area.

others roads' characteristics?

Response:

We appreciate your comment. Nevertheless, we prefer to keep these two factors as road density and road area density can reflect different dimensions. For example, although many road densities in old towns of Europe are high, their road area density is still low because of their narrow road width. Therefore, road density and road area density cannot be equivalent.

Comment 17: And what about car accessibility? Don't you think it could be significant to have accessibility also by car/bus/...

Response:

Many thanks for your question. Actually, we have involved the accessibility by cars, i.e., BTA5600 via sDNA analysis.

“•BTA5600:

As shown in Figure 6, the study areas have good average values of BTA5600, between 6269~100762.2. These successful cases reveal that there are some roads with high betweenness for vehicles in the study areas. Some of them are parallel or perpendicular and close to the coastline, and some are in the depth of the hinterlands and perpendicular or parallel to the coastline. These roads are also usually wider for vehicles and the parts of the city centers.”  (Section 4.1.1., Page 7, Line 251-257)

Comment 18: And the presence of a pedestrian road? or the parking spaces availability? They're different from your indicator "street accessibility"

Response:

Thanks for your reminder. We accepted that pedestrian roads are important analysis elements, but we have not gotten access to this data. Thus, we acknowledge our limitation in section 5. A detailed revision has been attached below.

“In addition, more features such as pedestrian roads and availability of public spaces may be related to urban vibrancy. We may take them into account in the future.” (Section 5.2., Page 20, Line 481-483)

Comment 19: missing a brief description of this technique

Response:

Thanks for your reminder. The description of the Spacematrix has been added in section 2.2.

“Moreover, Spacematrix invented by Berghauser Pont is a useful technique which can classify various kinds of building types quantitatively with a set of morphological indicators, e.g., GSI, and average floors [24].” (Section 2.2., Page 3, Line 111-113)

Comment 20: Of the renovated waterfront or in the selected area? maybe typology and intensity can be different in existing or newly transformed urban areas...

Response:

We acknowledged that the contexts, typologies, and intensities in the renovated area and the study areas are different, so it is not appropriate to apply the values of the cases to the values of the renovated area immediately. However, we still admire the common features and ranges of the indicators in the cases, which can guild the renovated area to some extent. In the future, to enhance credibility, we will add more categories of the cases to enhance the representation of the cases to guild urban morphology.

“More cases should be included in the future, and quantitative sampling methods should be attempted to bring more robust results.” (Section 5.2, Page 20, Line 478-480)

Comment 21: And what about population? For example, density, age, education, employment…

Response:

Thanks for pointing out this question. We agree that the dimensions of populations are essential. Firstly, the FAR has implied the number of populations in an area. Furthermore, this study mainly focuses on the indicators that can be extracted from urban morphology and related to urban vibrancy. We have to acknowledge it is not easy for urban design to control these social dimensions like age, education, and employment. However, we will try to involve extra aspects of the social dimensions in our future study.

“Fourthly, although current study mainly focuses on urban morphological indicators, we would involve the indicators from social dimensions in our future study to achieve a more systematic consideration.” (Section 5.2, Page 21, Line 487-489)

Comment 22: in which cases? generally, I don't think so...

Response:

We apologized that the previous version is unclear. We have replaced it with a clarified explanation.

“The blocks in or near the regional centers and compact roads have higher road density and road area density values.” (Section 4.1.1., Page 7, Line 227-228)

Comment 23: I can't understand the visualization: in blue the road density shown on building blocks? how it's calculated if the colored blocks are just buildings? or do the represented blocks include buildings and roads? if yes, in which way you decided to select some road to be put aside (in white in the figure)

Response:

Thanks for your question. Firstly, we apply blocks as basic spatial analysis units for assisting urban design that is often practiced based on block scale. Secondly, as to the indicators in the road aspect, we calculated the value by the road within the walking distance for each block. Explanations about this method have been added at the end of the 3.2 section.

“To provide insight into the urban design that is often practiced based on block scale, we have used the block as the calculation unit for each indicator except the indicator of betweenness. There are two reasons for this consideration. First, the block is widely regarded as an operation spatial unit when designers intervene in the urban form [47]. Second, the effectiveness of block has been verified by many empirical studies [14,15,34]. As to the indicators of road density, road area density, and road intersection density, we calculated their values by the road within the walking distance of 300 meters [48] for each block. However, the road segment has been used as the calculation unit for the indicator of betweenness. That is because this indicator mainly reflects the street network structure, thus we want to know which roads have higher values and have more opportunities to be the regional or city centers through the betweenness.” (Section 3.2., Page 7, Line 203-213)

Reference:

Zhang, A.; Li, W.; Wu, J.; Lin, J.; Chu, J.; Xia, C. How can the urban landscape affect urban vitality at the street block level? A case study of 15 metropolises in China. Environment and Planning B: Urban Analytics and City Science 2020, 48, 1245-1262, doi:10.1177/2399808320924425.

Comment 24: source

Response:

Thanks for your reminder. We have added a new reference.

Reference:

Chiaradia, A.; Hillier, B.; Schwander, C.; Wedderburn, M. Compositional and urban form effects on centers in Greater London, UK, Proceedings of the Institution of Civil Engineers - Urban Design and Planning 2012, 165, 21-42, doi:10.1680/udap.2012.165.1.21.

Comment 25: I see that the average values are very different, and also the range is different... how can you define, from this case study, an opportune range for the betweenness indicators?

Response:

Many thanks for your question. Firstly, the betweenness was mainly used to describe the street network structure and to find if there are any spatial centers in each case. Secondly, we acknowledged that although the average betweenness among benchmark cases has a big range, the difference between the problematic area and the benchmarks is bigger as shown in Figure 18.

Comment 26: source, or case studies, supporting the sentence

Response:

Thanks for your reminder. We have added a new reference.

“Generally, areas with high pedestrian choice become regional service centers more easily, and areas with high vehicle choice become city centers more easily [49].” (Section 4.1.1., Page 8, Line 238-239)

Reference:

Chiaradia, A.; Hillier, B.; Schwander, C.; Wedderburn, M. Compositional and urban form effects on centers in Greater London, UK, Proceedings of the Institution of Civil Engineers - Urban Design and Planning 2012, 165, 21-42, doi:10.1680/udap.2012.165.1.21.

Comment 27: Also, in this figure, road intersection density is shown on blocks. please explain why and what assumptions were made in order to do so.

Response:

Many thanks for your question. The reason has been added to the text and detailed revisions have been attached below.

“To provide insight into the urban design at the block scale, we have used the block as the calculation unit for each indicator except the indicator of betweenness. There are two reasons for this consideration. First, the block is widely regarded as an operation spatial unit when designers intervene in the urban form [47]. Second, the effectiveness of block has been verified by many empirical studies [14,15,34].” (Section 3.2., Page 7, Line 203-213)

Reference:

Zhang, A.; Li, W.; Wu, J.; Lin, J.; Chu, J.; Xia, C. How can the urban landscape affect urban vitality at the street block level? A case study of 15 metropolises in China. Environment and Planning B: Urban Analytics and City Science 2020, 48, 1245-1262, doi:10.1177/2399808320924425.

Comment 28: urban vitality

Response:

Thanks for your suggestions. We have added the words urban vibrancy into the sentences (Table 2).

Comment 29: equipped with public space or brownfield sites?

Response:

OSR is the ratio of the ground without cover to the total floor area, which can show the extent of average-available vacant land for each person for an area. The vacant land in the article refers to the land without coverage which includes public spaces, brownfields, and any land without coverage.

“The open space ratio (OSR) index represents the ratio of vacant land which means lands without coverage to floor area [41].” (Section 4.1.2., Page 13, Line 319-320)

Comment 30: it's not spatial as GIS intends... it's the distribution in the case studies.

Response:

We have deleted all the words “spatial” below the figures.

Comment 31: Average value is not statistically significant: other statistical variables are needed

Response:

Thanks for your comment. We have added a new indicator, standard deviation, to describe the degree to which data points are clustered around the mean (Table 3).

Comment 32: Is this one more similar to some of the case studies?

Response:

In this section, we use the same indicators to calculate the Jianshatan area and compare the results of the Jinshatan area with the benchmark value, in 4.2. section. The values of the Jinshatan area are different from the values of benchmarks.

Comment 33: Perfect... it seems to be really different, from this point of view…

Response:

Thanks for your admiration.

Comment 34: Please translate this figure.

Response:

We have replaced the wrong figure with the right one. Thanks for your advice (Figure 20).

Comment 35: Where? showing the range for each indicator?

Response:

Thanks for your reminder. Firstly, the value range of each indicator has been illustrated in section 4.1.4 and the table3 below.

“To be specific, in the street network configuration aspect, the road density, road area density, road intersection density, and block size are expected to be set between 14-33.5, 7-21, 86.3-533.3, and 73.9-143.2 respectively. Second, the FAR, GSI, OSR, and average floor are expected to be set between 0.63-3.1, 19-43, 3.3-8.2, and 30.6-127.9 respectively. Finally, the MXI and facility density are expected to be set between 30.6-127.9, 416.4-2687.4 respectively.” (Section 4.1.4, Page 17, Line 378-383)

Secondly, to avoid duplication, we only summarize the common characteristics of vibrant waterfronts in section 5. Detailed revisions have been attached below.

“To be specific, it can be found from the illustration in sections 4.1 and 4.2 that the vibrant waterfronts have compact street networks and building types that are conducive to public activities. They also possess high development intensity and mixed-dense land use.” (Section 5.1, Page 20, Line 451-454)

Comment 36: You need to refer to the study that identified them as benchmarks

Response:

“The features were extracted by the analysis of the morphological indicators related to vibrancy from representative benchmark cases which were selected from travel websites and under the strict requirements [25].” (Section 5.1., Page 20, Line 445-447)

Reference:

TripAdvisor. Available online: https://www.tripadvisor.com/TravelersChoice (accessed on 12 September 2022)

Comment 37: ref table/figure

Response:

“For example, based on the results of this study, the road density for a vibrant waterfront is suggested to be set in the range of 14.0-33.5 km/km2 (Figure 3), while the floor area ratio should be set in the range of 0.63-3.1 (Table 3).” (Section 5.1, Page 21, Line 458-460)

Comment 38: The selection methodology is not so clear, mainly in the first step of defining dimensions and indicators.

Response:

Thanks for this question. First, to make the introduction of the selection methodology clearer, the reference to the text has been added. The detailed revisions have been attached below.

“Firstly, for the methods of case selection, due to the difficulty of using quantitative methods, the benchmarks were selected by a qualitative approach based on the pre-defined conditions (section 3.1.1).” (Section 5.2, Page 21, Line 476-478)

As to the defining dimensions and indicators, the relevant details can be found in section 3.2 as below.

“Starting from the “Anglo-Germanic Historico-geographers” and “town-plan analysis” research methods proposed by Conzen [20], the urban form can be disassembled into town planning, typology models, and land use models. Town planning can be divided into street organizations, plots, blocks, and buildings. According to the definitions mentioned above, to summarize the relevant features, the quantitative urban morphology can be divided into three dimensions as shown in Table 1, including street network configuration, typology and development intensity, and land use. The meaning of each indicator has also been explained below.” (Section 3.2, Page 5, Line 174-181)

Again, we appreciate all your insightful comments. We would like to express our appreciation to the editor and anonymous reviewers in the acknowledgment for the time and effort you have taken to provide such insightful guidance. We would be glad to respond to any further questions and comments that you may have.

Round 2

Reviewer 2 Report

most of the comments generated a change in the text, making it clearer. the interpretation of the maps, often not so easily correlated with spatial values and relationships, leaves me feeling confused